# Modulation of translational decoding by m⁶A modification of mRNA

Sakshi Jain[1,5], Lukasz Koziej [2,5], Panagiotis Poulis [1,5], Igor Kaczmarczyk [2,3], Monika Gaik [2], Michal Rawski [2,4], Namit Ranjan[1], Sebastian Glatt [2] ✉ & Marina V. Rodnina [1] ✉

N⁶-methyladenosine (m⁶A) is an abundant, dynamic mRNA modification that regulates key steps of cellular mRNA metabolism. m⁶A in the mRNA coding regions inhibits translation elongation. Here, we show how m⁶A modulates decoding in the bacterial translation system using a combination of rapid kinetics, smFRET and single-particle cryo-EM. We show that, while the modification does not impair the initial binding of aminoacyl-tRNA to the ribosome, in the presence of m⁶A fewer ribosomes complete the decoding process due to the lower stability of the complexes and enhanced tRNA drop-off. The mRNA codon adopts a π-stacked codon conformation that is remodeled upon aminoacyl-tRNA binding. m⁶A does not exclude canonical codon-anticodon geometry, but favors alternative more dynamic conformations that are rejected by the ribosome. These results highlight how modifications outside the Watson-Crick edge can still interfere with codon-anticodon base pairing and complex recognition by the ribosome, thereby modulating the translational efficiency of modified mRNAs.

Post-transcriptional modifications of messenger RNA (mRNA) modulate key steps in mRNA metabolism, including mRNA stability and localization[1], nuclear export[2], exon-intron architecture[3], and alternative splicing[4]. N⁶-methyladenosine (m⁶A) is one of the most abundant internal mRNA modification[5] found in both coding and untranslated regions of mRNAs in all three domains of life with an average of three m⁶A per mRNA transcript[6,7]. The modification sites are located within a conserved consensus sequence of RRACH (R = A/G, H = U/A/C; UGCCAG in *Escherichia coli*), recognized by dedicated m⁶A methytransferases (writers) and demethylases (erasers)[8,9]. The activity of the catalytic subunit m⁶A-METTL writer complex (MAC) is stimulated by the regulatory subunit m⁶A-METTL-associated complex (MACOM)[10]. Reader proteins (e.g. eIF3, YTH domain proteins, HNRNP proteins in eukaryotes) recognize m⁶A and its consensus sequence, imparting a highly localized and regulated action[11,12]. The prevalence and conservation of m⁶A modification and of the machinery that

installs, reads, and removes the modification suggests its important role in regulating mRNA dynamics and gene expression at the post-transcriptional level. Indeed, recent studies link m⁶A modification to a wide range of important biological processes such as neural development and differentiation[13], spermatogenesis[14], cell growth and cancer[15]. However, in most cases the exact regulatory pathway is unknown.

The majority of m⁶A sites are mapped within mRNA coding regions[5,6,16], where the modification can modulate codon reading by aminoacyl-tRNA (aa-tRNA) in the A site of the ribosome. Most of the work on the mechanism by which m⁶A affects decoding has been carried out in the *E. coli* translation system; given the evolutionary conservation of the decoding mechanism[17,18], similar mechanisms are likely to operate in bacteria and eukaryotes. Although the modification is not located on the Watson−Crick edge of the nucleotide, it decreases translation efficiency[19], slows down the decoding process[20], and alters

[1]Max Planck Institute for Multidisciplinary Sciences, Göttingen 37077, Germany. [2]Malopolska Centre of Biotechnology, Jagiellonian University, Krakow 30-387, Poland. [3]Doctoral School of Exact and Natural Sciences, Jagiellonian University, Krakow 30-387, Poland. [4]National Synchrotron Radiation Centre SOLARIS, Jagiellonian University, Krakow 30-387, Poland. [5]These authors contributed equally: Sakshi Jain, Lukasz Koziej, Panagiotis Poulis. ✉e-mail: sebastian.glatt@uj.edu.pl; rodnina@mpinat.mpg.de

the ribosome selectivity for the cognate aa-tRNA (i.e. fully matching the codon)[20,21]. Single-molecule experiments using Förster Resonance Energy Transfer (smFRET) that measure the duration of the decoding step (up to and including the peptide bond formation step) and tRNA−mRNA translocation showed that decoding is slowed down by m⁶A modification at the first position of the AAA codon (m⁶AAA) (15-fold) and Am⁶AA (8-fold) and to a smaller extent by AAm⁶A, Cm⁶AG and CCm⁶A modification (2.5-fold)[20]. Rapid kinetics measurements of GTP hydrolysis suggested that m⁶A reduced the initial selection capacity of the ribosome[20,21]. In addition, m⁶AAA caused a 1.5-fold excess hydrolysis of GTP per peptidyl transfer reaction, indicating that m⁶A induces excessive proofreading of cognate aa-tRNA[20]. Extrapolations of the $k_{cat}/K_M$ values led the authors to suggest that m⁶A dramatically (10-fold) decreases the association rate ($k_a$) of the ternary complex EF-Tu−GTP−aa-tRNA (TC) to the ribosome prior to codon recognition[21]. The postulated effect would imply that the modification induces a conformational state of the ribosome that is refractory to TC binding; however, structural data in support of this hypothesis is lacking. Thus, while experiments clearly show that the m⁶A modification has an effect on translation, the exact mechanism of how m⁶A regulates individual steps of the decoding cycle remain elusive.

In this study, we systematically analyze the effect of m⁶A at all three AAA codon positions on different steps of the decoding cycle by Lys-tRNA^Lys using a combination of ensemble kinetics, smFRET, and single-particle cryo-EM techniques. In contrast to previous suggestions, we show that the rates of forward reactions of initial binding and codon recognition are largely unaffected, whereas the respective dissociation steps are faster in the presence of the modification, which explains the lower effective rate of GTP hydrolysis. Furthermore, the presence of m⁶A increases the transition time from codon recognition to post-decoding phase, which—together with an increased tRNA drop-

off – results in a significantly fewer ribosomes reaching the translocation step. We show that the m⁶A effect depends on the position, the nature of the codon and on the chemical modifications of the tRNA anticodon. High-resolution cryo-EM structures reveal how the ribosome accommodates m⁶A modifications at different codon positions and suggest that the AAA codon adopts a structured conformation in the A site independently of the modification. In summary, the most dramatic effect of m⁶A modification appears to involve destabilization of the codon-anticodon interaction, which results in tRNA drop-off at all stages of decoding and explains the inhibitory effect of the modification on translation.

## Results

### Effect of m⁶A on the elemental steps of decoding

We monitored the consecutive decoding steps of an AAA codon modified at any of the three codon position by its cognate Lys-tRNA^Lys using a fully-reconstituted in vitro translation system from *E. coli*[22]. Using the AAA codon enables to monitor the codon position effect of m⁶A within a single codon. In the first step of decoding (Fig. 1a), the ternary complex EF-Tu−GTP−Lys-tRNA^Lys (TC) binds to the ribosome in a codon-independent manner (characterized by the rate constants $k_1$ and $k_{-1}$), thereby initiating codon reading[17]. In the next step, codon recognition ($k_2$ and $k_{-2}$) induces domain closure of the SSU and triggers GTPase activation of EF-Tu ($k_3$) upon EF-Tu docking at the GTPase-activating center in the large ribosomal subunit (LSU), resulting in GTP hydrolysis ($k_{GTP}$)[17,23–25]. The release of the reaction product P_i ($k_4$;[26]) induces the transition of EF-Tu from the GTP- to the GDP-bound conformation, allowing aa-tRNA accommodation in the A site of the LSU ($k_5$) and peptide bond formation ($k_{pep}$), whereas EF-Tu is released from the ribosome ($k_6$). Alternatively, aa-tRNA can by rejected from the ribosome ($k_7$)[17]. Additional rejection steps, for example after GTP

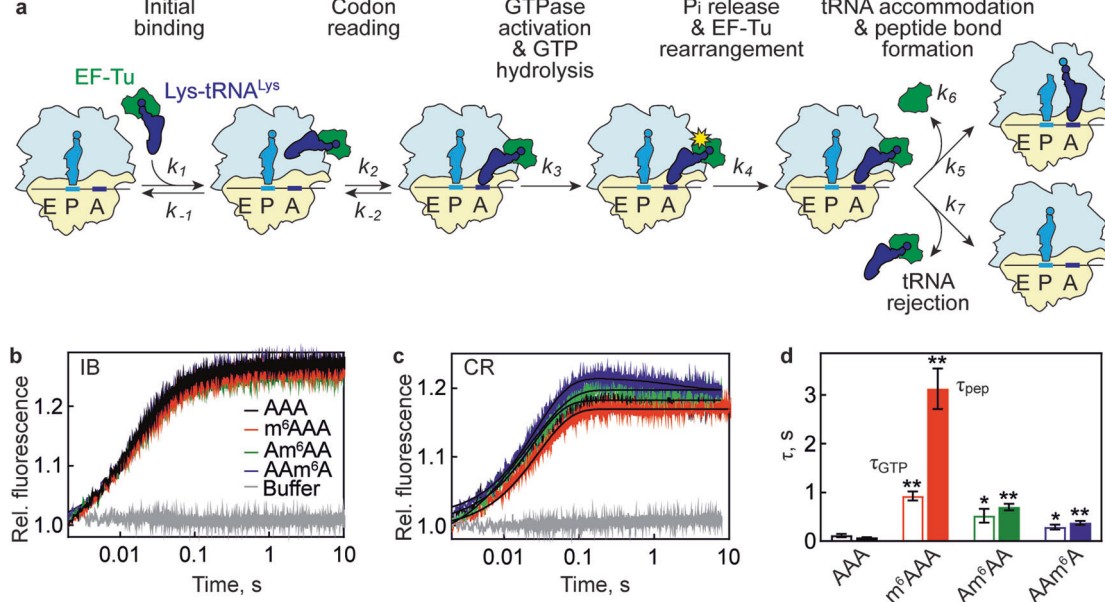

**Fig. 1 | Effect of m⁶A modification on the kinetics of decoding. a** Kinetic scheme of decoding. SSU, yellow; LSU, light blue. E, P, and A indicate the tRNA binding sites of the ribosome[22,27]. **b** Fluorescence change reporting codon-independent initial binding (IB) of TC to the ribosome. The A-site codon AAA is unmodified (black), m⁶AAA (red), Am⁶AA (green) or AAm⁶A (blue). Each time course is an average of 5-6 technical replicates. The $k_{app}$ values from exponential fitting are 70 ± 1 s⁻¹ (AAA), 75 ± 1 s⁻¹ (m⁶AAA), 65 ± 1 s⁻¹ (Am⁶AA), and 61 ± 1 s⁻¹ (AAm⁶A) for the major phase (80% of the fluorescence change). Control was obtained by mixing TC with buffer (gray). **c** Fluorescence change of EF-Tu−GTP−[¹⁴C]Lys-tRNA^Lys(Prf16/17) reporting A-site codon recognition (CR). Each time course is an average of 5-6 technical replicates. The $k_{app}$ values of the major reaction (fluorescence increase) are

33 ± 1 s⁻¹ (AAA), 25 ± 1 s⁻¹ (m⁶AAA), 32 ± 1 s⁻¹ (Am⁶AA), and 37 ± 1 s⁻¹ (AAm⁶A). Controls were obtained by mixing TC with buffer (gray). **d** Reaction times (τ) of GTP hydrolysis (τ_GTP, open bars) and peptide bond formation (τ_pep, closed bars) from exponential fitting of the respective time courses (Supplementary Fig. 1a). Error bars indicate SEM of the fit generated from three independent experiments (N = 3). *P* values are calculated using two-tailed unpaired Student's *t*-test in comparison to unmodified AAA codon (*P < 0.05, **P < 0.005). For τ_GTP, *P* = 0.0011 for m⁶AAA, *P* = 0.0496 for Am⁶AA and *P* = 0.0419 for AAm⁶A. For τ_pep, *P* = 0.0019 for m⁶AAA, *P* = 0.0008 for Am⁶AA and *P* = 0.0023 for AAm⁶A. For all panels of Fig. 1, TC (Lys or Phe) (0.3 µM) was mixed with IC (0.9 µM). Source data are provided as a Source Data file.

**a** Lys2: 5'- AUG-**AAA**-UUC-GUU-AC-3'
Lys4: 5'- AUG-GUG-UUC-**AAA**-CUC-3'

**b** Lys: 5'- AUG-**AAA**-UUC-GUU-3'
Thr: 5'- AUG-**ACC**-UUC-GUU-3'

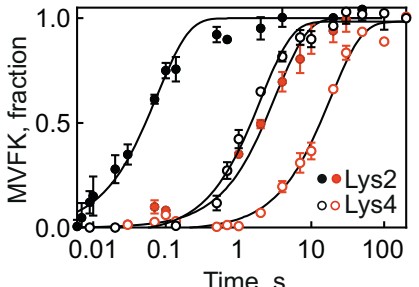
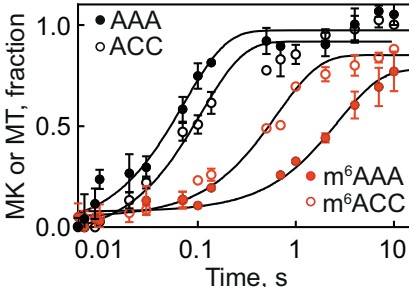

**Fig. 2 | m⁶A position in the mRNA and codon identity modulates extent of translation delay.** The coding sequences show the unmodified and first-position modified A-site AAA or ACC codon (bold). **a** Time courses of f[³H]Met-Val-Phe-[¹⁴C]Lys (Lys4 mRNA, open circles) and f[³H]Met-[¹⁴C]Lys (Lys2 mRNA, closed circles) formation obtained by mixing IC (0.9 µM) with the AAA (black) or m⁶AAA (red) with TC (0.3 µM) containing Val-tRNA^Val, Phe-tRNA^Phe, and [¹⁴C]Lys-tRNA^Lys and EF-G (4 µM) or [¹⁴C]Lys-tRNA^Lys respectively. Smooth black lines represent one-exponential fits. The reaction time τ calculated from exponential fitting is 0.08 s

for AAA vs 3.3 s for m⁶AAA at Lys2 and 2 s for AAA vs 18 s for m⁶AAA at Lys4. Thus, the delay in decoding due to the modification is -3 s at Lys2 and -16 s for Lys4. **b** Dipeptide formation with Lys TC or Thr TC (0.3 µM) upon binding to IC (0.9 µM). f[³H]Met-[¹⁴C]Lys (closed circles) or f[³H]Met-[¹⁴C]Thr (open circles) dipeptides were separated by HPLC and quantified by scintillation counting. For both panels of this figure, each time course is the mean of three independent experiments with error bars representing standard deviation (N = 3). Source data are provided as a Source Data file.

hydrolysis but prior to dissociation of aa-tRNA from EF-Tu are likely[24], but are accounted for in the global rejection rate $k_7$. Notably, all rejection rates are very low for the cognate Lys-tRNA^Lys ([22]), which results in the efficient decoding and peptide bond formation.

To quantify the m⁶A effect, we prepared ribosome initiation complexes (IC) with f[³H]Met-tRNA^fMet bound to the AUG codon in the P site and an AAA (or m⁶AAA, Am⁶AA, or AAm⁶A) codon in the A site. The IC was mixed with TC that contained reporters suitable to monitor each step of decoding using ensemble rapid kinetics[22,27]. To test the effect of m⁶A modification on the initial binding step prior to codon recognition, we used an established assay with a fluorescence reporter group proflavin attached to the tRNA^Phe elbow region (position 16/17)[28]. The reaction does not proceed beyond initial binding, because tRNA^Phe (anticodon AAG) does not match the AAA codon in the A site. Interaction of EF-Tu−GTP−Phe-tRNA^Phe(Prf16/17) with the IC leads to a rapid fluorescence increase that is largely independent of the mRNA modification, with the apparent rate constant $k_{app1}$ in the range of 60–75 s⁻¹ and very similar endpoints for all four codons (Fig. 1b). Although the apparent rate constant includes both the association ($k_1$[IC]) and dissociation ($k_{-1}$) terms, these results exclude the possibility that the association rate constant ($k_1$ or $k_a$) is decreased 10-fold by m⁶A modification as previously suggested[21]. Rather, the effect of the modification on the initial binding is small, if any.

Next, we monitored the codon recognition step using Lys-tRNA^Lys labeled by proflavin at positions 16/17 (tRNA^Lys(Prf16/17))[22]. After initial binding to the IC, Lys-tRNA^Lys recognizes its cognate AAA codon, leading to a fluorescence increase when EF-Tu−GTP−[¹⁴C]Lys-tRNA^Lys(Prf16/17) is mixed with IC (Fig. 1c). The apparent rate of codon recognition ($k_{app2}$) is essentially independent of the m⁶A modification, showing only a small variations from approximately 33 s⁻¹ with AAA to 25-37 s⁻¹ with m⁶A-containing codons and a minor effect on the amplitude of the signal change, consistent with a notion that the effect of m⁶A on initial binding and codon recognition is small. In contrast, GTP hydrolysis and peptide bond formation are strongly affected (shown as reaction times $\tau_{GTP}$ and $\tau_{pep}$ for better comparison; Fig. 1d and Supplementary Fig. 1a), in agreement with previous reports[20]. With m⁶AAA codon, $\tau_{GTP}$ was 8-fold and $\tau_{pep}$ was 40-fold higher than with AAA, indicating a significant slowing down of the decoding step. The effects of modifications at the 2nd and 3rd position were somewhat lower, decreasing $\tau_{GTP}$ by about 5-fold (Am⁶AA) or 2.4-fold (AAm⁶A) and $\tau_{pep}$ by 10-fold (Am⁶AA) or 5-fold (AAm⁶A) compared to unmodified AAA. We also tested the effect of the m⁶A modification

on translocation using tripeptide formation as a proxy for successful movement of the ribosome along the mRNA (Supplementary Fig. 1b). Ribosome complexes that completed decoding and contain tRNA^fMet in the P site and f[³H]Met-[¹⁴C]Lys-tRNA^Lys in the A site were mixed with EF-G to induce mRNA–tRNA translocation and a TC with Phe-tRNA^Phe that binds to the next codon resulting in synthesis of f[³H]Met-[¹⁴C]Lys-Phe (MKF) peptide. Comparison of the time courses of MKF synthesis shows that m⁶A at the 1st codon position slows down translocation by about 2-fold, whereas the 2nd- and 3rd-position modifications have no significant effect (Supplementary Fig. 1b).

## Context effects
To test whether the codon position in the mRNA attenuates the m⁶A effect, we compared the rates of AAA and m⁶AAA decoding at the 2nd (Lys2) and 4th (Lys4) positions in the mRNA. In the latter case, we translated a 4-codon sequence for Met-Val-Phe-Lys (MVFK) with unmodified AAA or m⁶AAA codon. We observe a 10-fold decrease in the $k_{pep}$ value of MVFK formation with m⁶AAA relative to AAA codon, compared to the 40-fold effect observed when the codons are in the 2nd position (Fig. 2a). The m⁶A effect on Lys4 is very similar to that observed by measuring ribosome rotation (15-fold)[20], which is a proxy for peptide bond formation. However, taking into account that incorporation of Lys4 is slower than of Lys2 due to additional time needed for MVF synthesis, the actual delay in decoding of the modified Lys4 codon (approximately 16 s; Fig. 2a) is even longer than on Lys2 (3 s; Fig. 1d). These results suggest that the m⁶A effect can be attenuated by the mRNA context, in agreement with the previous results[20].

m⁶A causes translation delays not only on modified AAA, but also on Cm⁶AG (coding for Gln) and CCm⁶A (Pro) codons[20], whereas the effect on the ACC (Thr) codon was not tested. Notably, ACC is part of the RRACH consensus sequence that is likely to be methylated in vivo. We measured the effect of m⁶A on the kinetics of dipeptide formation (MK or MT) with IC programmed by the respective mRNAs and TC (Lys or Thr). Our results show that the effect on the ACC codon is smaller than on AAA, 5-fold vs. 40-fold, respectively (Fig. 2b). Thus, also the codon/amino acid identity modulates the translation delay caused by the m⁶A modification.

## Interplay with tRNA modifications
In the cell, not only mRNA, but also tRNAs can be modified, which prompted us to test how modifications at the tRNA anticodon affect reading of m⁶A-modified codons. We made use of the known detailed

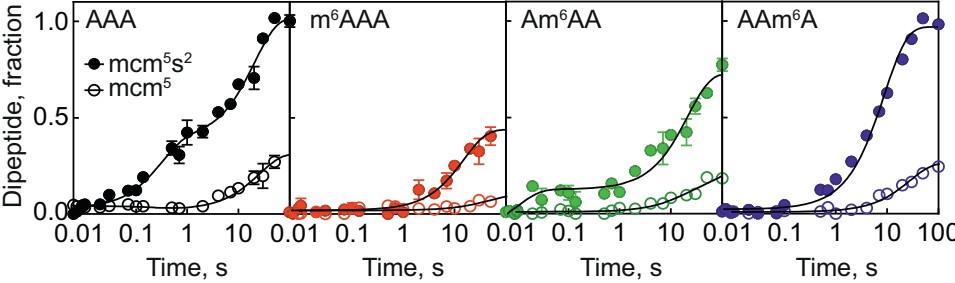

**Fig. 3 | Interplay between mRNA and tRNA modifications.** Time courses of fMet-[$^{14}$C]Lys formation obtained by mixing TC containing yeast [$^{14}$C]Lys-tRNA$^{Lys}$ (0.3 µM) with IC (0.9 µM). [$^{14}$C]Lys-tRNA$^{Lys}$ is fully modified with mcm$^5$s$^2$U$_{34}$ (closed circles) or hypomodified containing mcm$^5$U$_{34}$ (open circles). Each curve is the mean of three independent experiments with error bars representing standard deviation ($N$ = 3). Smooth black lines represent single or double exponential fits. fMet-[$^{14}$C]Lys dipeptides were separated by HPLC and quantified by scintillation counting. The $\tau_{pep}$ values of the fast reaction with fully modified mcm$^5$s$^2$U$_{34}$ [$^{14}$C]Lys-tRNA$^{Lys}$ are approximately 10–20 s$^{-1}$ irrespective of the modification, the reactions with mcm$^5$U$_{34}$ tRNA are too inefficient to obtain reliable values. Source data are provided as a Source Data file.

kinetic mechanism of AAA decoding by yeast tRNA$^{Lys}$ containing mcm$^5$s$^2$ modification at U$_{34}$ and the well-documented contribution of the s$^2$U$_{34}$ to decoding[22]. The mcm$^5$s$^2$ modification at U$_{34}$ is introduced by the *ELP*[29] and *URM1* pathways[30]. Deletion of the *URM1* gene in yeast results in synthesis of Δs$^2$U$_{34}$ tRNA$^{Lys}$, which carries all other modifications but lacks s$^2$U$_{34}$[31], as verified by ((N-acrylolamino)phenyl) mercuric chloride (APM) gel retardation of tRNA (Methods). The hypomodified tRNA shows an increased propensity to dissociate from the ribosome (manifested in the increased $k_{-2}$ and $k_7$ dissociation rate constants), slower rearrangement after GTP hydrolysis ($k_4$) and slower accommodation into the peptidyl transferase center ($k_5$)[22]. To test how a fully modified mcm$^5$s$^2$ Lys-tRNA$^{Lys}$ and Δs$^2$U$_{34}$ Lys-tRNA$^{Lys}$ affect m$^6$A decoding, we monitored fMet-[$^{14}$C]Lys formation with IC containing a unmodified or m$^6$-modified AAA codon in the A site (Fig. 3). The time course of peptide bond formation with the fully modified mcm$^5$s$^2$ Lys-tRNA$^{Lys}$ on the AAA codon is biphasic (Fig. 3a), which reflects heterogeneous pathways towards aa-tRNA accommodation depending on the timing of its release from EF-Tu ($k_{5a}$ and $k_{5b}$, respectively)[22]. Removal of the s$^2$U$_{34}$ modification dramatically reduces the rate and end level of the reaction (Fig. 3), in agreement with the earlier report[22]. With m$^6$A modification at any codon position, the rate of peptide bond formation with mcm$^5$s$^2$ Lys-tRNA$^{Lys}$ is very low, 30–80-fold lower than on an unmodified AAA codon (Fig. 3). Furthermore, the end level is reduced, indicating that a large fraction of Lys-tRNA$^{Lys}$ dissociates from the ribosome before entering the reaction. The effect is even more dramatic when the m$^6$A modification on the mRNA is combined with the Δs$^2$U$_{34}$ Lys-tRNA$^{Lys}$, in particular on m$^6$AAA and Am$^6$AA codons. These data suggest that m$^6$A modification enhances rejection of mcm$^5$s$^2$ Lys-tRNA$^{Lys}$ and the lack of s$^2$U$_{34}$ tRNA modification further destabilizes the complexes, thereby practically abolishing translation.

**The mechanism of decoding inhibition by m$^6$A**
While there is a clear effect of m$^6$ modification on GTP hydrolysis and dipeptide formation, the model explaining these effects by a slower association rate of the TC to the ribosome[21] is clearly not supported by our experimental data. This prompted us to examine the mechanism of decoding in more detail using smFRET approaches[32–35] comparing AAA and m$^6$AAA codons, which gave the largest rate differences. smFRET experiments are designed to monitor binding and unbinding of TC to and from the ribosome, as well as movement of aa-tRNA through the ribosome, accommodation and the dynamics of the peptidyl-tRNA after peptide bond formation[32,33]. TC binding to the ribosome was monitored by smFRET between the ribosomal protein L11 labeled with FRET donor Cy3 (L11-Cy3) and Lys-tRNA$^{Lys}$ labeled with FRET acceptor Cy5 (Lys-tRNA$^{Lys}$(Cy5)) (Fig. 4a). L11-Cy3 IC was immobilized on coverslips through 5′-biotinylated mRNAs carrying AAA or

## Table 1 | Kinetic analysis of decoding of AAA and m$^6$AAA codons

| | AAA | m$^6$AAA |
|---|---|---|
| $k_{-1}$[a], s$^{-1}$, ($n$)[b] | 8.1[c] (90) | 15 ± 1[d] (495) |
| $k_{CR\rightarrow forward}$, s$^{-1}$ ($n$) | 6.3 ± 0.7 (261) | 7.2 ± 2.2 (256) |
| $k_{-2}$[e], s$^{-1}$, ($n$) | 0.34 ± 0.02 (161)[f] | 1.1 ± 0.1 (95)[f] |
| $k_{CR\rightarrow A/P^*}$, s$^{-1}$ ($n$) | 2.0 ± 0.9 (138) | 1.0 ± 0.6 (125) |
| $k_{0.8\rightarrow 0.6}$, s$^{-1}$ ($n$) | 2.9 ± 0.6 (433) | 2.8 ± 0.2 (710) |
| $k_{0.6\rightarrow 0.8}$, s$^{-1}$ ($n$) | 4.4 ± 0.4 (441) | 4.2 ± 0.2 (701) |

[a]Shown are the corrected rates $k_{corrected}$ according to $k_{corrected} = k_{observed} - k_{photobleach} - 1/T$, where $k_{observed}$ the rate of the single-exponential decay function, $k_{photobleach} = 0.03 ± 0.01$ s$^{-1}$ and $T$ the observation window, 33 s[35].
[b]$n$, total number of transitions from three independent experiments.
[c]Number of traces is too small to calculate the rate in the independent datasets.
[d]Shown are the mean and standard deviation from 3 independent experiments.
[e]From the experiment with EF-Tu(H84A).
[f]The difference in the rates is statistically significant ($P$ = 0.0052).

m$^6$AAA codons in the A site. After injection of TC with Lys-tRNA$^{Lys}$(Cy5) into the flow chamber, we observed four types of traces. The percentages of ribosomes following each of type of trajectories are strikingly different for AAA and m$^6$AAA (Fig. 4b). Traces that show rapid (>30 s$^{-1}$) appearance of FRET signal followed by a rapid loss of FRET reflect initial binding complex formation and dissociation prior to codon recognition (Fig. 4b–d and Supplementary Fig. 2a). Such traces comprise only a small percentage of complexes on the AAA codon (8 ± 2%), but increase to 37 ± 1% ($P$ = 0.0015) on m$^6$AAA. The dissociation rate ($k_{-1}$) of these traces is higher with m$^6$AAA, indicating that m$^6$A promotes rejection of ternary complexes from the initial binding complex (Fig. 4b, Supplementary Fig. 2b and Table 1).

Two other types of traces correspond to ribosome complexes where tRNA is successfully accommodated in the A site. In those traces that show a full decoding trajectory, rapid (>30 s$^{-1}$) codon reading results in appearance of high FRET 0.9, followed by the progression to lower FRET states (step assignment is from[33]). The latter corresponds to tRNA movement from codon reading towards the accommodation in the A site and peptide bond formation. After peptide bond formation, fMet-Lys-tRNA$^{Lys}$(Cy5) starts to fluctuate between A/A, A/P (FRET 0.8) and A/P$^*$ (FRET 0.6) states (Fig. 4e, f and Supplementary Fig. 2c). The percentage and the decay rate of the codon reading state ($k_{CR\rightarrow forward}$) are not strongly affected by the m$^6$ modification (Fig. 4b, Supplementary Fig. 2d and Table 1), suggesting that the codon reading state does not accumulate during the reaction. However, the transition rate $k_{CR\rightarrow A/P^*}$ from the codon reading (FRET 0.9) to the post-decoding A/P$^*$ state (FRET 0.6) is reduced 2-fold in the presence of m$^6$A (Supplementary Fig. 2c, d and Table 1).

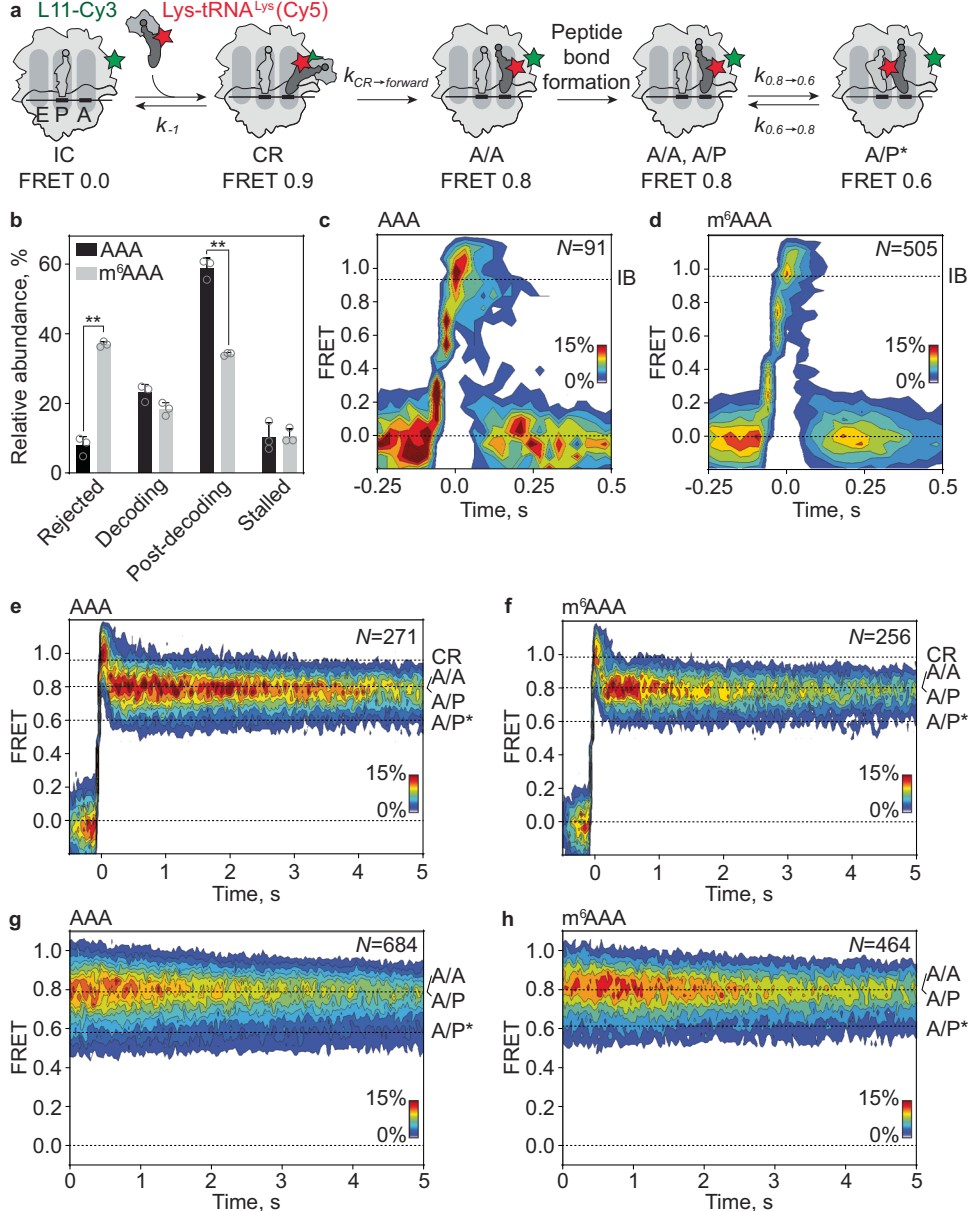

**Fig. 4 | Decoding of AAA and m⁶AAA monitored by smFRET. a** Schematic of the smFRET experiment. 70S IC (light gray) is labeled with Cy3 (green star) at the protein L11 and Lys-tRNA$^{Lys}$ is labeled with Cy5 (red star). **b** Relative abundance of the different types of traces during decoding of AAA (black) or m⁶AAA (gray) codon. "Rejected" denotes the fraction of complexes that dissociate from the ribosome after initial binding (IB; **c**, **d**). "Decoding" pertains to traces that show the characteristic codon reading (CR) phase followed by accommodation and ensuing fluctuations between A/A, A/P, and A/P* states (**e**, **f**). "Post-decoding" are the complexes in which Lys-tRNA$^{Lys}$ is found in an A/A, A/P or A/P* state at the start of the recording (**g**, **h**). "Stalled" denotes the fraction of ribosomes that do not progress past the CR state (Supplementary Fig. 2). Shown are the mean values with error bars representing standard deviation from $N = 3$ independent experiments. ** indicates statistical significance with $P < 0.01$ using a two-tailed Welch's t-test without adjustment for multiple comparisons. $P = 0.0015$ for rejected traces, $P = 0.0038$ for post-decoding traces, $P = 0.0565$ for decoding traces. Open

circles are the values in the individual experiments. **c** Contour plot showing the distribution of FRET values after synchronization during initial binding and dissociation of Lys-tRNA$^{Lys}$(Cy5) from ICs carrying the AAA codon in the A site. Traces are synchronized to the first transition with FRET > 0. $N$, number of traces. **d** same as **c** on the m⁶AAA codon. **e** Contour plot showing the distribution of FRET values during the full decoding trajectory on the AAA codon. FRET $0.96 \pm 0.03$ in the CR state derived from mean ± s.d. of the μ values of the Gaussian fitting of $N = 3$ independent experiments. **f** same as **e** on the m⁶AAA codon, FRET $0.98 \pm 0.01$. **g** Contour plot showing the distribution of FRET values in the post-decoding complexes carrying fMet-Lys-tRNA$^{Lys}$(Cy5) on the AAA codon. FRET population distribution reveals two states with FRET $0.79 \pm 0.01$ (A/A, A/P) and $0.58 \pm 0.02$ (A/P*)[35,64–66]. **h** Same as **g** on m⁶AAA codon. A/A, A/P, FRET $0.80 \pm 0.01$; A/P*, FRET $0.61 \pm 0.01$. Data are from N = 3 independent experiments. Source data are provided as a Source Data file.

A significant fraction of ribosome complexes reached the post-decoding state already at the starting point of imaging, with fMet-Lys-tRNA$^{Lys}$(Cy5) fluctuating between FRET 0.8 (A/A, A/P) and 0.6 (A/P*) states (Fig. 4g, h and Supplementary Fig. 2e). Here, we observe no significant differences in the transition rates between A/A, A/P (FRET 0.8) and A/P* (FRET 0.6) states ($k_{0.8 \to 0.6}$ and $k_{0.6 \to 0.8}$) in the presence of

m⁶A, indicating that m⁶A has no effect on the tRNA$^{Lys}$ dynamics in classical and hybrid states (Supplementary Fig. 2f and Table 1). However, the percentage of ribosomes showing classical to hybrid fluctuations is reduced in the presence of m⁶A ($P = 0.0038$, Fig. 4b), consistent with the notion that a fraction of ribosomes has lost the tRNA before entering the post-decoding phase. Lastly, we also

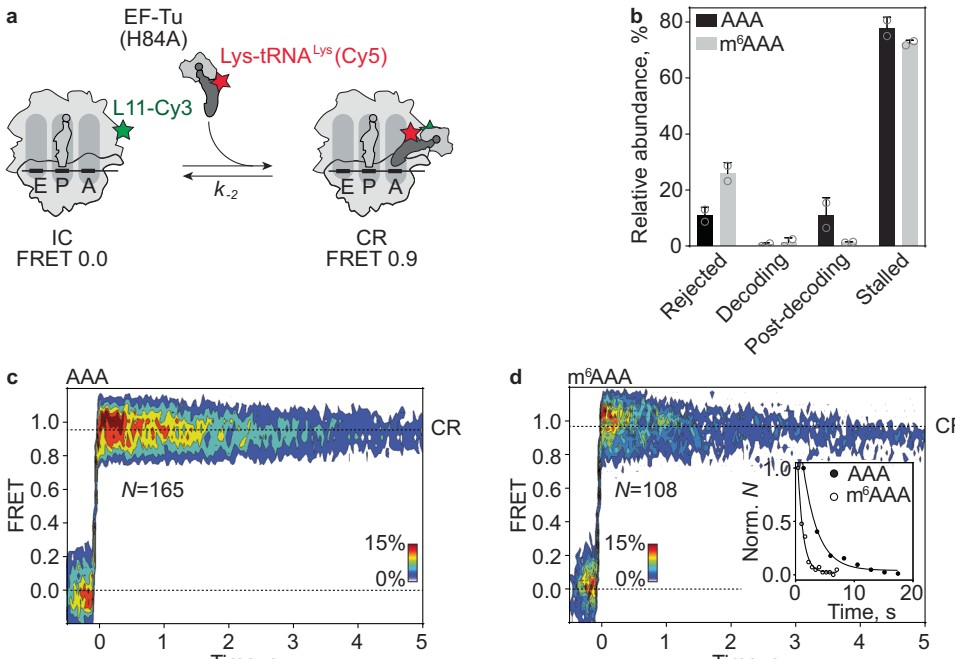

**Fig. 5 | Codon reading of AAA and m⁶AAA by EF-Tu(H84A)–GTP–Lys-tRNA^Lys(Cy5). a** Schematic of the smFRET experiment monitoring decoding attempts by the GTPase-deficient TC. CR, codon reading. **b** Relative abundance of smFRET traces during decoding of AAA (black) or m⁶AAA (gray) codons by EF-Tu(H84A). Shown are the mean values and error bars represent standard deviation from N = 2 independent experiments. Open circles are the values in the individual experiments. **c** Contour plot showing the distribution of FRET values after

synchronization during CR (FRET 0.97 ± 0.01) without progression to accommodation of Lys-tRNA^Lys(Cy5) on AAA codon. Data are from N = 2 independent experiments. **d** Same as **c** for the m⁶AAA codon (FRET 0.96 ± 0.01). Inset shows normalized dwell-time histograms of EF-Tu(H84A)–GTP–Lys-tRNA^Lys(Cy5) dissociation ($k_{-2}$) during decoding of AAA (closed circles) and m⁶AAA (open circles) codon (Table 1). Normalization was performed by division by the maximum value of the histogram. Source data are provided as a Source Data file.

observed a small fraction of traces that did not progress beyond the codon reading state, but the percentage of these traces was not affected by the presence of m⁶A (Fig. 4b and Supplementary Fig. 2g–i). In conclusion, m⁶A increases the dissociation of TC during initial binding and codon reading, as well as delays the transition to the post-decoding state after GTP hydrolysis.

To further investigate the effect of m⁶A on the stability of codon-anticodon duplex in the A site, we repeated the experiments using EF-Tu(H84A), a GTPase-deficient EF-Tu mutant (Fig. 5a). Steps prior to GTP hydrolysis, i.e. initial binding, codon recognition and GTPase activation, are not affected by the mutation, but GTP hydrolysis and the subsequent steps of tRNA accommodation and peptide bond formation are abolished[36]. In the presence of EF-Tu(H84A), the TC stalled in the codon reading state represents the majority of the population (78 ± 3% for AAA and 72 ± 1% for m⁶AAA) (Fig. 5b), in agreement with previous studies using non-hydrolysable GTP analogs[20,33]. For both AAA and m⁶AAA, FRET population distribution revealed a single FRET 0.9 codon reading state that did not progress towards accommodation (Fig. 5c,d). The decay rate of the long-lived codon reading state reports the dissociation of the ternary complex from the ribosome and acts as a proxy for codon-anticodon stability[22,27]. In the presence of m⁶A, the decay rate ($k_{-2}$) is 3.2-fold higher that on the unmodified AAA codon (Fig. 5d, inset), indicating that m⁶A destabilizes the codon-anticodon interactions during codon reading prior to GTP hydrolysis. This leads to rapid dissociation from the ribosome and thus further accounts for low efficiency of tRNA accommodation.

## Cryo-EM captures dipeptidyl-tRNA^Lys bound to m⁶A-modified codons

Next, we obtained the structures of ribosome complexes with tRNA^Lys in the A site using single-particle cryo-EM. The complexes were

prepared by mixing EF-Tu–GTP–Lys-tRNA^Lys with IC containing fMet-tRNA^fMet in the P site and mRNAs with either AAA, m⁶AAA, Am⁶AA, or AAm⁶A in the A site. We collected four equivalent datasets (Supplementary Table 1) and sorted the particles according to the tRNA occupancy of the E, P, and A sites (Supplementary Fig. 3). In addition to a fraction of unreacted IC, we found ribosomes that completed the decoding process and contained tRNA^fMet in the P site and fMet-Lys-tRNA^Lys in the A site. The latter complexes could be further classified into (i) those where tRNAs were in classical (P/P, A/A) states with ribosomes in a non-rotated conformation and (ii) those with tRNAs in hybrid (P/E, A/P) states with the ribosomal subunit rotated relative to each other. The particle distribution was dramatically different between samples with unmodified AAA and m⁶A-modified codons (Fig. 6a). Specifically, 48% (m⁶AAA), 52% (Am⁶AA), or 46% (AAm⁶A) of ribosomes failed to progress from the IC towards decoding and peptide bond formation, compared to only 4% of ICs with unmodified AAA codon. These differences support the notion that a sizable fraction of Lys-tRNA^Lys that attempts to bind to its cognate codon is rejected during the decoding process.

The particles with tRNAs in classical states were used to reconstruct maps at 2.3 Å (AAA), 2.9 Å (m⁶AAA), 2.8 Å (Am⁶AA), and 2.6 Å (AAm⁶A) global resolutions (GSFSC with 0.143 cut-off; Fig. 6a), respectively. The high quality of the maps in the proximity of the decoding region (Supplementary Fig. 4) allowed the direct visualization of the m⁶A modification at different codon positions (Supplementary Fig. 5). Furthermore, the fMet-Lys moieties in the peptidyl transferase center (Supplementary Fig. 6), as well as all known tRNA^fMet and tRNA^Lys modifications (e.g. $Cm_{32}$, $mnm^5U_{34}$) could be assigned. However, detailed comparisons did not reveal any pronounced structural differences between the models with and without m⁶A (Fig. 6b). Of note, we positioned the m⁶A in the *anti* conformation, which enables hydrogen bonding in all three codon-anticodon pairs.

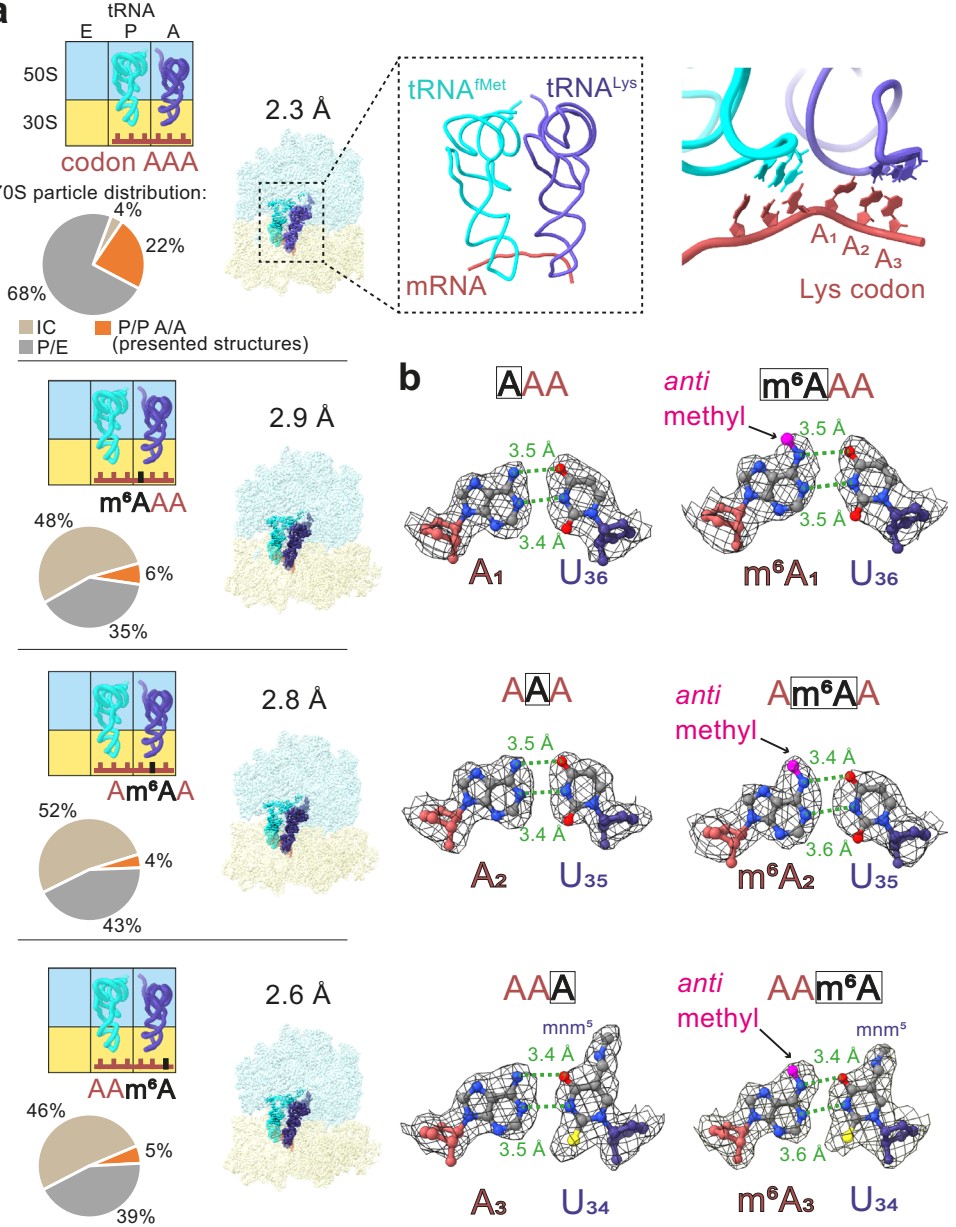

**Fig. 6 | Cryo-EM analysis of ribosome complexes with m⁶A modification at different codon positions.** **a** From top to bottom: AAA, m⁶AAA, Am⁶AA, AAm⁶A (top left: schematic view, right: map colored by the RNA chain). LSU is shown in light blue, SSU in yellow, mRNA in red, P-site tRNA in cyan and A-site tRNA in dark blue). The pie charts show relative particle distributions of different states. Lys codons carrying an m⁶A modification lead to a lower proportion of ribosomes progressing towards decoding (orange slice). The maps are shown as surface representation. Top middle panel, close-up view of the P-site and A-site tRNAs and mRNA. Top right panel, close-up view of mRNA–tRNA interactions. **b** Base pairing at the 1st, 2nd and 3rd codon position. Left, AAA codon; right, the respective m⁶A-modified pair. The m⁶ methyl group (magenta) is shown in an *anti* conformation. The measured distances between the conventional hydrogen bond donor and acceptor indicated with a dashed line are expressed in Å. The maps are shown as mesh with contour rmsLevel 4.

Even at the high resolution obtained, the cryo-EM densities do not unambiguously exclude the *syn* conformation. However, fitting m⁶A in a *syn* conformation precludes hydrogen bonding and results in an increased distance of the N6 atom to the O4 atom from the cognate U in the codon by ~0.5 Å (Supplementary Fig. 7). Thus, once the A-site aa-tRNA is bound, we cannot detect any major structural differences between unmodified and m⁶A-modified codons.

### The AAA codon is structured in an unoccupied A site

To further elucidate the structural basis of slower translational rates of ribosomes decoding the m⁶A-modified Lys codon, we separately analyzed ICs from the same datasets and reconstructed maps at 2.4 Å (m⁶AAA), 2.2 Å (Am⁶AA), and 2.1 Å (AAm⁶A) overall resolution (Supplementary Fig. 8). Because the dataset for the IC with unmodified AAA codon was too small to obtain a structure at a similar resolution range (Fig. 7a), we collected two additional IC datasets with either an AAA or AAm⁶A codon in the A site (Fig. 7b). The complexes were prepared in the same way as described above, but without adding Lys-tRNA^Lys; the final overall resolution of both structures was 2.0 Å (Supplementary Fig. 9 and 10). Despite the high quality of the maps, particularly in the decoding region (with local resolution between 2.4–2.7 Å), the density for the m⁶ modification was not well resolved (Supplementary Fig. 11). Surprisingly, in all structures the AAA codon was in a π-stacking conformation (Fig. 7c), which—to our knowledge—was not identified with any other codon sequence in the well-resolved IC structures available so far. The π-stacking was stabilized by the

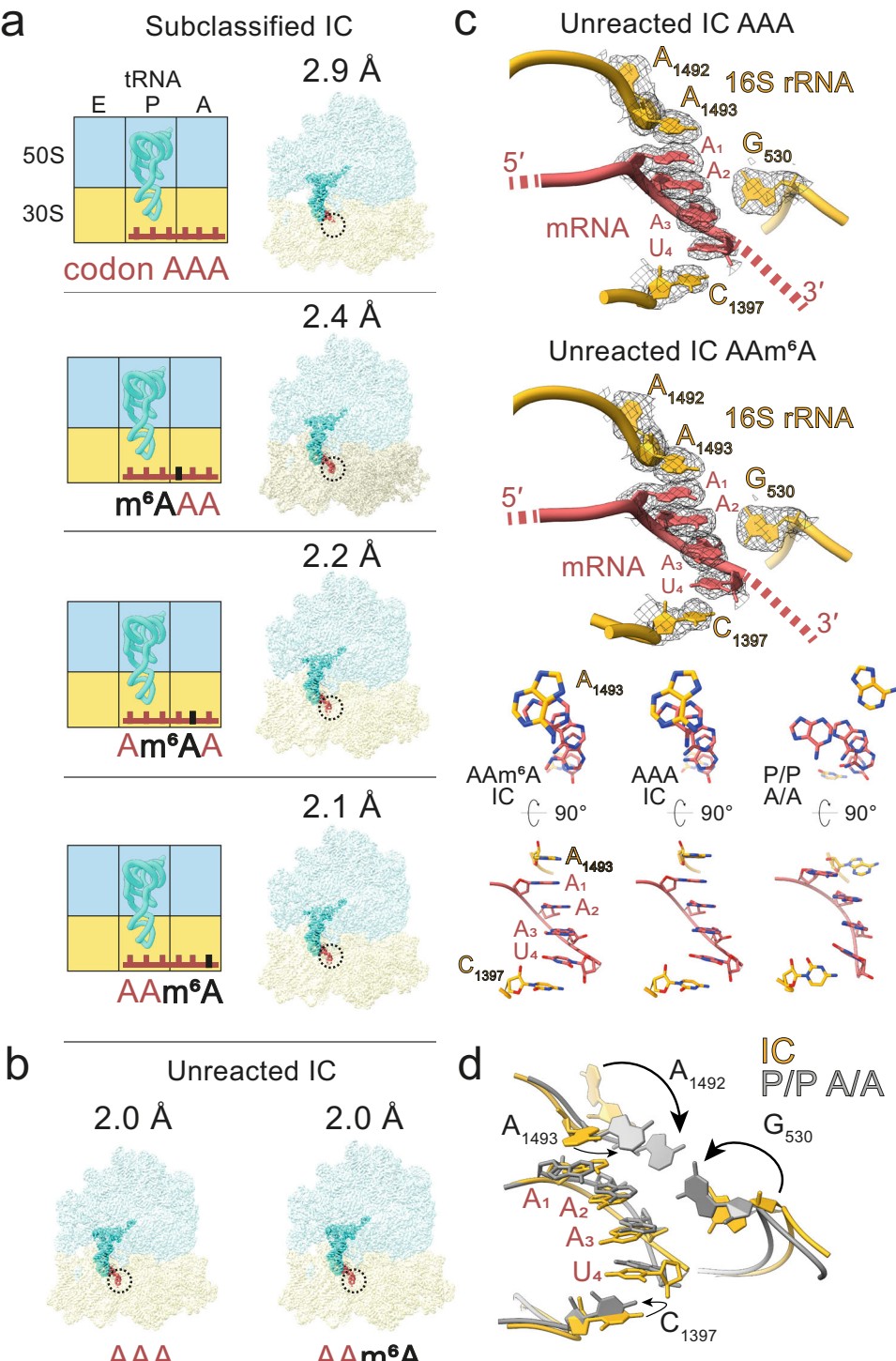

**Fig. 7 | Cryo-EM structures of ICs with m⁶A in different positions of the A-site codon. a** From top to bottom: AAA, m⁶AAA, Am⁶AA, AAm⁶A. Left panel: schematic view; right panel: map colored by the RNA chain. LSU is shown in blue, SSU in yellow, mRNA in red, P-site tRNA in cyan. The A-site codon is indicated with dashed circles. **b** Structures of unreacted ICs (i.e. prepared without addition of Lys-tRNA^Lys, see text) with AAA mRNA (left) and Am⁶AA mRNA (right). Maps are shown in surface representation. **c** Close-up view of interactions between AAA or AAm⁶A A-site codon (red), and rRNA (yellow) in the decoding region of the unreacted ICs (top). The corresponding models are also shown from different perspectives (bottom). In the unreacted AAA (left) and AAm⁶A ICs (middle) the nucleotides form a π-stacking array, whereas the equivalent nucleotides in the post-decoding complex (P/P A/A) appear unstacked (right). **d** Repositioning of the 16S rRNA nucleotides A1493, A1492, G530 and C1397. The conformational rearrangements during the transition from IC (yellow) to the post-decoding complex (P/P A/A; gray) are shown in reference to the mRNA codon.

interactions with the decoding center of the ribosome, in particular 16S rRNA nucleotides A1493 and C1397. Residue A1493 in helix 44 (h44) flips out of its helical arrangement and is involved in stacking with A1 of the AAA codon. The stacking array extends beyond the A-site codon to the next nucleotide ($U_4$) from the downstream codon UUC and with C1397 of 16S rRNA. Another key residue in the decoding center, A1492, remains stacked within an internal loop of h44 of the IC, whereas G530 is pushed away from the mRNA and does not appear to be engaged in the codon stabilization (Fig. 7c).

Comparison between the unpaired AAA codon in IC and the post-decoding complex with fMet-Lys-tRNA$^{Lys}$ in the A site shows that the positioning of the 16S rRNA nucleotides C1397, A1492, A1493, and G530 changes between the two states (Fig. 7d). Upon decoding, the base array of $A_1$-$U_4$ mRNA becomes unstacked. The C1397 base rotates by approximately 90° with respect to the $U_4$. While, in the IC, A1493 is involved in the stabilization of the A-site codon, in the decoding complex, A1492, A1493, and G530 monitor the correct Watson–Crick geometry of the codon-anticodon complex in the A site, consistent with previous findings[37,38]. A1493 and A1492 are flipped-out of the 16S rRNA stacking conformation and together with G530 interact with the minor grove of the codon-anticodon mini helix. Of note, the positions of these rRNA and mRNA nucleotides are identical in the m⁶A-modified and the AAA datasets (Fig. 7c), showing that m⁶A does not affect monitoring of the codon-anticodon complex by the ribosome.

## Discussion

The results of this work, together with previous reports[20,21], show that m⁶A modification on AAA and other A-containing codons, such as Glu, Gln, Pro, and Thr, leads to inefficient decoding in a codon context-dependent manner. We show that, in contrast to previous suggestions[21], m⁶A does not affect the association rate of the TC to the ribosome and provide detailed insights into how m⁶A modifications modulate the decoding process during translation elongation. Prior to decoding, the AAA codon adopts a π-stacking arrangement involving the 3′-adjacent U in the mRNA and stabilized by A1493 and C1397 of 16S rRNA (Fig. 7). While the codon-independent initial binding of the cognate TC-Lys to the AAA-programmed ribosomes is rapid and independent of the m⁶A modification (Figs. 1 and 3), the resulting complexes tend to dissociate more rapidly when the modification is present (Table 1; Figs. 4 and 5). As a result, fewer ribosomes proceed towards rapid codon recognition (Fig. 4). The presence of m⁶A reduces the stability of the codon recognition complex (Table 1) and slows down GTP hydrolysis and accommodation compared to the unmodified codon (Fig. 1 and refs. 20,21). The delays in the forward decoding steps together with the increased tRNA$^{Lys}$ drop-off from the ribosome result in fewer ribosomes that complete decoding (Figs. 4 and 6). On those ribosomes that reached the post-decoding state, the π-stacking arrangement of the codon is resolved, and the codon-anticodon complex adopts a canonical Watson-Crick geometry stabilized by the residues of 16S rRNA independent of m⁶A modification (Fig. 6).

Comparison of the structural and biophysical data provides insights into the dynamics of AAA decoding and the role of the m⁶A modification. The observed π-stacking codon arrangement is similar to that found in yeast ribosomes and is stabilized by homologous nucleotides (Supplementary Fig. 12 and refs. 39,40), indicating evolutionary conservation of the recognition mechanism. However, our data show that a single unmodified AAA codon is efficiently decoded despite the unfavorable codon conformation (Figs. 1 and 6), indicating that the structuring of the A-site codon alone cannot explain translational stalling on extended poly(A) tracks, as previously suggested[39,40]. Also the presence of an AAAA sequence in the yeast structure instead of an AAAU in our structures cannot explain the difference, as AAAAAA sequences are translated efficiently[39]. Concerning the effect of the m⁶A modification, previous NMR data suggested that m⁶A stabilizes stacking compared to the unmodified base[41]. A stronger codon

stacking in a conformation unfavorable for codon reading may explain why a sizable fraction of TCs that scan the ribosome during the initial binding and codon reading are rapidly rejected from the ribosome prior to forming a codon-anticodon complex, as indicated by smFRET experiments (Fig. 4b–d).

The rates of the codon recognition on those ribosomes that proceed towards decoding are not much changed by the m⁶A modification (Fig. 1c). This observation favors the conformational selection model, which assumes that the π-stacking and unstacked arrangements are in dynamic equilibrium on the ribosome. If the TC attempting to read the A site codon encounters the stacked conformation, the respective attempt is likely to fail, resulting in the dissociation of the TC prior to codon recognition. In contrast, when TC encounters an unstacked conformation, it can rapidly form a codon-anticodon complex and proceed towards tRNA accommodation. At this stage, also the switch from *syn* to *anti* conformation of the m⁶ group may play a role, as the optimal codon-anticodon pairing is accomplished by rotating away from its energetically preferred *syn* geometry on the Watson-Crick face to the higher-energy *anti* conformation[41]. The m⁶ modification is poorly resolved in the cryo-EM structures with the vacant A site, despite the high quality of the 70S IC structures (Fig. 7a, b). Given that the nucleotides are clearly resolved, the weak density for the modification (Supplementary Fig. 11) likely reflects its rotational dynamics. Notably, the density for the additional methyl group is better resolved in the post-decoding complexes (Supplementary Fig. 5) indicating that tRNA binding to the codon restricts m⁶ dynamics. Fitting the density with m⁶A in *anti* conformation yielded a perfect Watson–Crick geometry of base pairing (Supplementary Fig. 7), whereas using the *syn* conformation results in a shift of the entire nucleotide. The unfavorable *anti* conformation is partially compensated by hydrogen bonding and Watson–Crick pairing with U resulting in the net destabilization by 0.5–1.7 kcal/mol for a single methyl substitution[42]. The fluctuations between *syn* and *anti* conformation of m⁶A in combination with the steric hindrance of the codon-anticodon pairing in the energetically more favorable *syn* conformation may cause the destabilization of the codon-anticodon complex observed in smFRET measurements (Table 1). The altered dynamics of the codon-anticodon complex may also affect the propensity of the SSU to form a closed global conformation that is essential for the activation of GTP hydrolysis[25,43,44], which may provide a potential mechanistic explanation for the observed slowing down of GTP hydrolysis.

The accommodation of tRNA$^{Lys}$ is also affected by m⁶A, in particular for the m⁶AAA codon, which is manifested by a further delay of peptide bond formation after GTP hydrolysis (Fig. 1d). smFRET experiments reveal the tendency for a delayed entry into the FRET 0.6 state upon the formation of tRNA hybrid states after peptide bond formation (Table 1). These data suggest that m⁶A enriches for conformations that enhance tRNA dissociation and disfavor the forward steps towards completion of the decoding step. However, the complexes that completed decoding are found in the canonical tRNA conformation regardless of the modification (Fig. 6). Thus, the ribosome rectifies the codon-anticodon dynamics to select the correct codon-anticodon geometry and the m⁶A modification affects this process, potentially by stabilizing otherwise unfavorable codon-anticodon conformations. Similarly, the ribosome selectively stabilizes G-U mismatches in a tautomeric form that is unfavorable in solution, but is favored by the ribosome, because it allows pairing with Watson–Crick geometry[42,45], thus underscoring the generality of the ribosome rectification mechanism.

The dynamics of the codon-anticodon complex is further modulated by the modifications in the tRNA anticodon (Fig. 3). With yeast tRNA$^{Lys}$, the presence of m⁶A had a strong effect not only on the rate of peptide bond formation, but also on the end level of reaction, which indicates enhanced drop-off of tRNA$^{Lys}$ from the ribosome during

decoding. Similarly to the reaction with *E. coli* tRNA$^{Lys}$, the most dramatic effect was observed with m$^6$AAA and the smallest with AAm$^6$A. Removal of the s$^2$ modification from yeast mcm$^5$s$^2$U$_{34}$ essentially inhibits the reaction (Fig. 3), underscoring the complex interplay between the dynamics of the codon and the properties of the tRNA anticodon for the outcome of tRNA selection. Taken together, our work combining ensemble kinetics, smFRET and cryo-EM methods provides detailed mechanistic insights into how m$^6$A modification modulates decoding.

## Methods

70S ribosomes, EF-Tu and f[$^3$H]Met-tRNA$^{fMet}$ from *E. coli* were prepared as described in previous studies[27,46,47]. Unmodified and m$^6$A-modified mRNA was purchased from IDT (Iowa, USA). Native Lys-tRNA$^{Lys}$, Lys-tRNA$^{Lys}$(Prf16/17) and Phe-tRNA$^{Phe}$ from *E. coli* and yeast Phe-tRNA$^{Phe}$(Prf16/17) were prepared as described[22,48]. Yeast mcm$^5$s$^2$U$_{34}$ and mcm$^5$U$_{34}$ Lys-tRNA$^{Lys}$ were prepared from wild-type S288C and $\Delta$*URM1* yeast strains, respectively[31], and the s$^2$ modification was quantified as described[22]. The extent of s$^2$ modification was verified by ((N-acrylo-lamino)phenyl)mercuric chloride (APM) gel retardation of tRNA and was 70–90% for native tRNA$^{Lys}$ and 2% for *urm1*$\Delta$ tRNA$^{Lys}$. The following mRNAs (unmodified and m$^6$A-modified at different codon positions of AAA or ACC codon) were purchased from IDT (Iowa, USA). The coding region (starting with AUG) is separated from the 5′UTR by space and the m$^6$ residue is introduced in one of the underlined A positions.

5′-GGCAAGGAGGUAAAUA AUGA̲A̲AUUCGUUAC-3′
5′-GGCAAGGAGGUAAAUA AUGA̲CCUUCCGCCUCUCUCUC-3′
5′-GGCAAGGAGGUAAAUA AUGGUGUUCA̲AACUGCGCCUCUCU CUC-3′

For the smFRET experiments, the following 5′-biotinylated mRNA (unmodified and m$^6$A-modified) was used.

5′-Biotin-CAACCUAAAACUUACACACCCGGCAAGGAGGUAAAUA AUGA̲AAUUC AUUACCUAA-3′

EF-Tu–GTP–[$^{14}$C]Lys-tRNA$^{Lys}$ (Prf16/17) (TC-Lys) was prepared by incubating EF-Tu (75 μM), GTP (1 mM), PEP (3 mM), DTT (1 mM), pyruvate kinase (1% v/v), tRNA$^{Lys}$(Prf16/17) (15 μM), ATP (3 mM), L-[$^{14}$C] lysine (22.5 μM) and Lys-tRNA synthetase (2% v/v) in buffer A containing 50 mM Tris-HCl pH 7.5, 70 mM NH$_4$Cl, 7 mM MgCl$_2$ and 1 mM DTT. Assembled TC-Lys was purified by gel filtration on two Superdex 75 HR columns operated in tandem (GE Healthcare) in buffer A. EF-Tu–GTP–[$^{14}$C]Phe-tRNA$^{Phe}$(Prf16/17) and EF-Tu–[γ-$^{32}$P]GTP–[$^{14}$C]Lys-tRNA$^{Lys}$ were prepared and purified in similar ways.

IC was prepared from 70S ribosomes (2 μM), mRNAs as indicated (6 μM), f[$^3$H]Met-tRNA$^{fMet}$ (3 μM), IF1, IF2, IF3 (3 μM each), DTT (1 mM) and GTP (1 mM) in buffer TAKM$_7$ (50 mM Tris-HCl pH 7.5, 30 mM KCl, 70 mM NH$_4$Cl and 7 mM MgCl$_2$) for 30 min at 37 °C. IC was purified by centrifugation through 1.1 M sucrose cushion in TAKM$_7$ for 2 h at 4 °C and 259,000 × *g* in Beckman Optima Max-XP ultracentrifuge. After centrifugation, the pellets were dissolved in TAKM$_7$ and quantified by scintillation counting.

### Rapid kinetics

All fluorescence stopped-flow experiments were performed in buffer A at 20 °C. Prf fluorescence was excited at 463 nm and measured after passing through a KV500 long pass filter (Schott). Experiments were performed by mixing equal volumes of IC (0.9 μM) with TC-Lys or TC-Phe (0.3 μM) as indicated and monitoring the time course of fluorescence change. Relative fluorescence was calculated by division of all fluorescence values by the value at time 0.

To monitor GTP hydrolysis, peptide bond formation, and mRNA–tRNA translocation following peptide bond formation, equal volumes of IC (0.9 μM) and respective TC (0.3 μM) were mixed in a quench-flow apparatus for time ranging from milliseconds to seconds in TAKM$_7$ at 37 °C. For GTP hydrolysis, the reaction was quenched with 50% formic acid. Intact [γ-$^{32}$P]GTP and γ-$^{32}$Pi were separated by TLC in

0.5 M KH$_2$PO$_4$[27]. The TLC plates were analyzed using phosphorimaging in Typhoon FLA9500 (GE Healthcare). For peptide bond formation and translocation, reactions at specific time points were quenched with 0.5 M KOH and the peptides released by alkaline hydrolysis at 37 °C. Peptide samples were neutralized by adding one fifth volume of 100% glacial acetic acid and analyzed by reversed-phase HPLC (LiChroSpher 100 RP-8 HPLC column, Merck) using 0–65% acetonitrile gradient in 0.1% trifluoroacetic acid. The HPLC fractions were quantified by double label radioactivity counting[49]. The data were normalized to an interval from 0 to 1 and exponential fitting of the data was performed with GraphPad Prism.

### smFRET experiments

Ribosomes labeled at protein L11 and Lys-tRNA$^{Lys}$ labeled with Cy5 at the 3-amino-3-carboxypropyl group at uridine 47 were prepared as described[32,35]. IC was prepared by incubating 70S(L11-Cy3) with a 3-fold excess of IFs, mRNA and f[$^3$H]Met-tRNA$^{fMet}$ and GTP (1 mM) for 30 min at 37 °C and purified by centrifugation through sucrose cushion (1.1 M) in TAKM$_{21}$ (50 mM Tris-HCl pH 7.5, 70 mM NH$_4$Cl, 30 mM KCl, 21 mM MgCl$_2$). The pellet was dissolved in TAKM$_7$. TC was prepared by incubating 3-fold excess EF-Tu (or EF-Tu(H84A) mutant) with GTP (1 mM), phosphoenolpyruvate (3 mM), and pyruvate kinase (0.5%) for 15 min at 37 °C and subsequent addition of Lys-tRNA$^{Lys}$(Cy5).

Biotin-coated glass objective slides and coverslips were prepared as described[35]. Reaction chambers were incubated with TAKM$_7$ containing putrescine (8 mM), spermidine (1 mM), BSA (10 mg/ml) and neutravidin (1 μM; Thermo Scientific) for 5 min at room temperature. Neutravidin was washed by adding the same buffer containing putrescine (8 mM), spermidine (1 mM) and BSA (1 mg/ml). Purified IC(L11-Cy3) were diluted to 1 nM in TAKM$_7$ containing putrescine (8 mM), spermidine (1 mM) and added to the reaction chambers for 2 min at room temperature. Imaging started after addition of TAKM$_7$ containing putrescine (8 mM), spermidine (1 mM), protocatechuic acid (2.5 mM), 50 nM *Pseudomonas* protocatechuate-3,4-dioxygenase (50 nM), 6-hydroxy-2,5,7,8-tetramethylchromane-2-carboxylic acid (1 mM) methylviologen (1 mM; Sigma-Aldrich), GTP (1 mM) and TC(Lys-tRNA$^{Lys}$(Cy5)) (5 nM).

TIRF imaging was performed at 22 °C on an IX 81 inverted microscope using a PLAPON 60 × 1.45 numerical aperture objective (Olympus). Cy3 was excited using a 561 nm solid-state laser operated at 25 mW and images were recorded with an electron multiplying CCD (charge-coupled device) camera (CCD-C9100-13, Hamamatsu) at a rate of 30.3 frames/s. Color channels were separated by projecting donor and acceptor emission on different parts of the CCD chip using an image splitter (dual view micro imager DV2, Photometrics), filter specifications HQ 605/40, HQ 680/30 (Chroma Technology).

Fluorescence time courses for Cy3 and Cy5 were extracted using custom-made MATLAB (MathWorks) software according to published protocols[32,35]. A semi-automated algorithm (MATLAB) was used to select single fluorophores showing anticorrelated fluorescence intensities and single-step photobleaching. Cy3 bleed-through into the Cy5 channel was corrected using an experimentally determined coefficient of 0.13. FRET efficiency was calculated as the ratio of the measured emission fluorescence intensities, FI$_{Cy5}$/(FI$_{Cy3}$ + FI$_{Cy5}$). Trajectories were truncated to remove photobleaching and photoblinking events. The set of all FRET traces for a given complex was compiled in a histogram, which was fitted to a sum of Gaussian functions. MATLAB code using an unconstrained nonlinear minimization procedure (fminsearch, MATLAB, R2011b) yields mean values and standard deviation for the distribution of FRET states. Two-dimensional contour plots were generated from raw time-resolved FRET trajectories using a custom-made software. smFRET trajectories were fitted by Hidden Markov model using the vbFRET software package (http://vbfret.sourceforge.net/)[50] to generate the idealized trajectories. FRET changes in idealized trajectories that were smaller than the s.d. of the Gaussian distribution of

the FRET states were not considered transitions because they could not be not distinguished from the noise. Dwell times of different FRET states were calculated from idealized trajectories. The dwell-time distribution was fitted to an exponential function, $y = y_0 + Ae^{-t/\tau}$ to calculate the decay rate ($k = 1/\tau$). GraphPad prism 8 software was used for the representation of smFRET data and fits of the data.

## Cryo-EM grid preparation

IC was prepared and purified as described above except that non-radioactive fMet-tRNA$^{fMet}$ was used. TC was prepared from EF-Tu (1.5 μM), EF-Ts (0.02 mM), Lys-tRNA$^{Lys}$ (0.3 μM), GTP (1 mM), phosphoenolpyruvate (3 mM), pyruvate kinase (1%), DTT (1 mM) in buffer TAKM$_7$. IC and TC were mixed in a 1:1 molar ratio, diluted to the final concentration of approximately $A_{260}$-20 (absorbance at 260 nm 10 mm path) and $A_{280}$-10 (absorbance at 280 nm 10 mm path), and incubated on ice for 60 s. The "unreacted" IC complexes contained all the above mentioned components except for Lys-tRNA$^{Lys}$. Approximately 3 μL of the solution was applied onto freshly glow-discharged TEM grids (Quantifoil R2/1, Cu 200 mesh) and plunge-frozen into liquid ethane by a Vitrobot Mark IV (Thermo Fisher Scientific) using the following parameters: humidity 100%, temperature 4 °C, blot total 1, wait time 0, blot force 0, blot time 2 s, drain time 0 s.

## Cryo-EM single-particle reconstruction

Cryo-EM datasets were collected at National Cryo-EM Centre SOLARIS (Kraków, Poland). The datasets of IC + TC complexes with AAA, m$^6$AAA, Am$^6$AA, and AAm$^6$A codons in the A site, as well as AAA unreacted-IC and AAm$^6$A unreacted-IC contained 8435, 9366, 7172, 7207, 10115, and 10276 movies, respectively (40 frames each). The movies were acquired using Titan Krios G3i microscope (Thermo Fisher Scientific) operated at 300 kV accelerating voltage, magnification of 105k, and corresponding pixel size of 0.86 Å/px. A K3 direct electron detector used for data collection was fitted with BioQuantum Imaging Filter (Gatan) using 20 eV slit. The K3 detector was operated in a counting mode. Imaged areas were exposed to 40 e$^-$/Å$^2$ total dose (corresponding to -16 e$^-$/px/s dose rate measured in vacuum). The frame stacks were obtained using under-focus optical conditions with a defocus range of −2.1 to −0.9 μm and 0.3 μm steps. The collected datasets were analyzed with cryoSPARC v3.3.0[51]. Firstly, patch motion correction and patch CTF estimation were performed. Next, approximately 500 particles were picked manually. The acquired sets of particles were subjected to 2D classification and used in the generation of preliminary classes for template picking. The application of a template picker and 2D classification of the datasets resulted in correspondingly 689123, 1113602, 732439, 774481, 1413299, and 1282719 particles, respectively (Supplementary Figs. 3 and 9). These sets were used for Heterogenous Refinement and particles from the classes corresponding to well-defined 70 S ribosomes served as an input for focused 3D classifications. The 3D Classification was performed in cryoSPARC using soft masks corresponding to E-, P- and A-site tRNA generated in Relion 3.1[52]. The unreacted 70S IC particles did not have any discernible density in the E and A sites, as expected for the IC. In the absence of Lys-tRNA$^{Lys}$, the fraction of particles with ordered P-site fMet-tRNA$^{fMet}$ reached 78% (AAA) and 80% (AAm$^6$A) in the two datasets, demonstrating the comparable quality of IC preparations. The final particle stacks were unbinned for local motion correction[53] in cryoSPARC. During final 3D Homogenous Refinements, the particles were subjected to Defocus Refinement, Global CTF Refinement, and Ewald Sphere Correction to generate high-resolution maps. Local map resolution was calculated using cryoSPARC. Prior to model fitting, the combined half-maps were sharpened with DeepEMhancer[54].

## Molecular modeling

The initial atomic model was assembled by combining core elements of *E. coli* 70S ribosome structures solved by cryo-EM at 2.0 Å (PDB: 7K00[55]) and 2.54 Å resolution (PDB: 6XZB[56]). The initial model of tRNA$^{fMet}$ was sourced from a 3.2 Å ribosome cryo-EM structure (PDB: 6WDD[24]) and tRNA$^{Lys}$ was isolated from a 3.6 Å ribosome cryo-EM reconstruction (PDB: 5JTE[57]). Other structural elements, such as mRNA, were built manually into the map using Coot[58]. Following rigid body fitting using ChimeraX[59], the atomic coordinates were flexibly fit with Namdinator[60] and Isolde[61]. The models were real-space refined in Phenix[62]. The final coordinates were validated using MolProbity[63] and the model statistics are presented in Supplementary Table 1. The cryo-EM maps and atomic models were displayed using ChimeraX version 1.2.5.

## Reporting summary

Further information on research design is available in the Nature Portfolio Reporting Summary linked to this article.

## Data availability

The micrographs, cryo-EM densities, and atomic models generated in this study have been deposited in the Electron Microscopy Public Image Archive (EMPIAR), the Electron Microscopy Data Bank (EMDB), and the Protein Data Bank (PDB) under the following accession codes: (1) AAA EMPIAR-11287; AAA IC EMD-16031 and PDB ID 8BGH; AAA P/P A/A EMD-16015 and PDB ID 8BF7. (2) m$^6$AAA EMPIAR-11290; m$^6$AAA IC EMD-16065 and PDB ID 8BHP; m$^6$AAA P/P A/A EMD-16062 and PDB ID 8BHN. (3) Am$^6$AA EMPIAR-11289; Am$^6$AA IC EMD 16059 and PDB ID 8BHL; Am$^6$AA P/P A/A EMD-16057 and PDB ID 8BHJ. (4) AAm$^6$A EMPIAR-11288; AAm$^6$A IC EMD-16029 and PDB ID 8BGE; AAm$^6$A P/P A/A EMD-16047 and PDB ID 8BH4. (5) AAA unreacted-IC EMPIAR-11291, EMD-16081 and PDB ID 8BIL. (6) AAm$^6$A unreacted-IC EMPIAR-11292, EMD-16082 and PDB ID 8BIM. Ensemble kinetics data are provided in the Source Data file. Processed smFRET data are provided in the Source Data file. Original images of smFRET experiments are available upon request due to their large size and the lack of a relevant public database. Source data are provided with this paper.

## Code availability

The codes used to analyze data in this study are available from the corresponding authors upon request.

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

## Acknowledgements

We thank M. Jaciuk, G. Ważny and P. Indyka for helpful discussions and O. Geintzer, V. Herold, T. Hübner, F. Hummel, S. Kappler, M. Klein, C. Kothe, A. Pfeiffer, T. Steiger and M. Zimmermann for expert technical assistance. The work was supported by the German Research Council (Deutsche Forschungsgemeinschaft) (Priority Programme SPP1784 to N.R. and M.V.R., Germany's Excellence Strategy - EXC 2067/1-390729940 and Leibniz Prize to M.V.R.) and the European Research Council (ERC) (grant agreement No 101001394- tRNAslation to S.G.). We thank the MCB structural biology core facility (supported by the TEAM TECH CORE FACILITY/2017-4/6 grant from Foundation for Polish Science, S.G.) for providing instruments and support. This publication was developed under the provision of the Polish Ministry of Education and Science project: "*Support for research and development with the use of research infrastructure of the National Synchrotron Radiation Centre SOLARIS*" under contract nr 1/SOL/2021/2. We acknowledge SOLARIS Centre for the access to the Krios microscope, where the measurements were performed. This research was supported in part by PLGrid Infrastructure (Academic Computer Centre Cyfronet AGH).

## Author contributions

S.J. performed biochemical and rapid kinetics ensemble experiments and data analysis. P.P. designed and performed smFRET experiments and analyzed the data. L.K. prepared most cryo-EM samples, analyzed the datasets and build atomic models with the support of I.K. and S.G. L.K., I.K. and S.G. analyzed the structures and prepared the structural figures. M.G. prepared initial cryo-EM samples, collected datasets and analyzed the data. M.R. collected all cryo-EM datasets, performed initial quality assessment and organized data storage. N.R., S.G. and M.V.R. conceptualized and supervised the research. S.J., L.K., P.P., S.G. and M.V.R. wrote the manuscript with contributions of all authors.

## Funding

## Competing interests

The authors declare no competing interests.
