## [Peer Review File · Nature Communications]

REVIEWER COMMENTS

Reviewer #1 (Remarks to the Author):

In their study, Jain et al. present compelling evidence that m6A destabilizes decoding complexes and encourages alternative conformations rejected by the ribosome, consequently impacting the translation efficiency of modified mRNAs. The authors hypothesize that the remodeling of pi-stacking confirmation upon tRNA binding may be hindered by RNA modifications. While I am not an expert in kinetics or translation mechanisms, I am confident in commenting on the cryo-EM data, which appears to be of high quality. As such, I recommend publication, with a focus on this aspect. To clarify the paper for a general audience, I propose the following suggestions:

1. The authors should more clearly differentiate between bacterial and eukaryotic translation. The introduction mentions eukaryotic facts, but the work is exclusively on prokaryotic ribosomes. It is important to prevent readers from mistakenly applying knowledge gained here to eukaryotes. The title and abstract should also explicitly state that this is prokaryotic research.
2. To account for Nature Communication's general audience, the manuscript should be more accessible. The authors should provide more detailed explanations of their approaches, the rationale behind each step, and how to interpret the results, rather than just referring to previous publications.
3. The paper employs a simpler sorting scheme than what was previously published by the Rodnina and Stark labs, which accounted for different ratcheting positions to obtain precise conformational distributions. It is surprising that ratcheting appears to have a minor role here. Moreover, the approach may be biased and not up to the current standards of the ribosome structural field, as some particles result in "low resolution maps." The authors should consider using a more advanced approach.
4. The pi-stacking and conformational change are crucial to the manuscript, but their presentation in the figure is difficult to appreciate. I suggest providing more density, a closer view, and a different viewing angle to better understand and evaluate the claims.

Minor issue:

5. The authors frequently use overall resolutions to argue map quality. However, this can be misleading, as the overall resolution of ribosomes is generally good, particularly for prokaryotic ribosomes. The study discusses local details, so local resolutions should be mentioned in the main text. Extended Data

Figure 4 could be improved by enlarging the encircled base densities and reducing the ribosome size to emphasize the relevant features.

Reviewer #2 (Remarks to the Author):

This manuscript from Jain S. and colleagues aims at identifying which translocation step is impacted by the presence of a m6A modification. To answer this question the authors used a combination of smFRET, cryoEM and stop-flow rapid kinetics in a fully reconstituted translation system. They identified that m6A has a strong impact on the acceptance rate of the tRNA leading to a quick dissociation of the tRNA.

This is a fascinating use of complementary techniques (smFRET, cryoEM, rapid kinetics) to address this question. This leads to a complete overview of the defects caused by the modification depending on its relative position within the codon.

The experiments have been carefully done, and are very well interpreted, the discussion is very interesting and clearly recapitulate the data obtained during this work.

My main concern is about the model they used. Indeed, they used a AAA codon (Lys) and ACC (Thr) to address the question of the role of m6A at the first position of the codon. They found there is a strong difference between both codons (ACC being 8-fold less impacted than AAA by the presence of the modification see fig2B) but they only rely on the literature for the other codons (although their data do not fit with previously published data on the role of m6A) with a A at the second and third position, such as gln (CAG) and pro (CCA).

AAA is a very special codon, as stated by the authors, due to its stacking conformation in absence of tRNA. Although this happens in various organisms like yeast, it is not clear for me if this is true for all codons or not.

This leads me to question the generalizability of these data. This is not really discussed in the manuscript, and I feel this could be a real improvement to perform the kinetics analysis with such codons. This is not a negligible question, because m6A is mainly found in coding sequences, so if m6A has a so strong effect on translation, it is very surprising that this does not leads to more translational defects, especially in cancer cells where m6A homeostasis is strongly modified.

I have few more comments for the authors:

- It would help the reader to better understand the experimental setup if the mRNA sequence used was indicated in the various figures. This is especially true for the context effect, where two lysine codons are positioned at position 2 and 4 of the mRNA. Do the differences observed between both codons due to the nucleotide located immediately downstream? This position effect is very curious, and the authors do not comment on that.

- In the first experiments it is not clear if the lysine tRNA used is fully modified or not. As this is a fully reconstituted translation system, it will be useful to clearly indicate modification status of the tRNA. Do the authors verify that after the isolation and the labelling tRNAs were still modified?

- Figure 7 is not completely clear. I guess we are looking ribosomes with an empty A site as suggested by the panel A? So why the panel D shows a post-decoding structure with a A-site tRNA?

Reviewer #3 (Remarks to the Author):

This manuscript from a leading laboratory in studies of protein synthesis mechanisms contains several interesting results well worthy of publication, combining ensemble, single molecule, and cryoEM studies in attempting to get a complete picture of the effects of m6A mRNA modification on translation. However, it is poorly presented and should be rejected in its current form.

MAJOR CRITICISMS

1. There is a consistent lack of rigorous, quantitative reasoning, which detracts from the significance of the results presented. In principle, the ensemble and single molecule results should reinforce one another, both qualitatively and quantitatively, but such direct comparisons are lacking. As one example, there is close to 100% of product formed for the AAA and m6AAA ensemble experiments (Fig. 2), but only more partial formations in the single molecule experiments (Fig. 4b) and this apparent discrepancy is not explained, nor are the extremely low fractions of product formation found in the cryoEM experiments (Fig. 6a).

Other examples of a lack of rigorous, quantitative reasoning:

p.6 – para 1 - penultimate sentence. What is the actual estimated delay for Lys 4 , and how was it quantified?

p. 7 – lines 5 – 8. The biphasic pathway. How was it shown that Lys-tRNA was fully modified? What other explanations of heterogeneity are possible? Why are m6AAA and Aam6A apparently monophasic.

p. 16 – lines 8, 9 from bottom – Is the energy difference observed in the NMR study sufficient to account for the differences observed in this manuscript?

2. Differences are cited with the work of Puglisi et al., but no attempt is made to explain possible reasons for these differences.

3. UNJUSTIFIED CONCLUSIONS.

p. 18 – lines 1-5. The conclusion that the difference in mcm5s2U34 vs. mnm5s2U34 in yeast vs. *E. coli* tRNAs accounts for the effects on peptide bond formation is totally unjustified, since other differences in sequence and modification of the two tRNAs (lines 1 and 2) could also be important. Indeed, the comments about biphasic kinetics that follow, lines 3-5, invoke such differences.

p. 17 – lines 3 – 5 from the bottom. Ribosome rectification of codon-anticodon dynamics. This conclusion is more strongly worded than is justified by the results. What other possibilities were considered?

OTHER CONCERNS

1. p. 4, Fig 1d – There is an apparent error in the results presented for AAA, since τ GTPase must be $>$ τ peptide.
2. p. 14 –The last sentence appears to be in direct contradiction of the major thesis of the manuscript.
3. The ensemble rate data, Figs. 1-3, should be summarized in a Table, for easy reference by the reader.
4. Rate units should be used consistently for both ensemble and sm data. The switch within the MS between k (s⁻¹) and τ (s), as in Fig. 1, is needlessly confusing.

MINOR POINT

p.19 – line 3. The tRNAs should be identified as coming from *E. coli*.

REVIEWER COMMENTS

Reviewer #1 (Remarks to the Author):

In their study, Jain et al. present compelling evidence that m⁶A destabilizes decoding complexes and encourages alternative conformations rejected by the ribosome, consequently impacting the translation efficiency of modified mRNAs. The authors hypothesize that the remodeling of pi-stacking confirmation upon tRNA binding may be hindered by RNA modifications. While I am not an expert in kinetics or translation mechanisms, I am confident in commenting on the cryo-EM data, which appears to be of high quality. As such, I recommend publication, with a focus on this aspect. To clarify the paper for a general audience, I propose the following suggestions:

Reply: We thank the reviewer for the positive comments and useful suggestions to improve the manuscript.

1. The authors should more clearly differentiate between bacterial and eukaryotic translation. The introduction mentions eukaryotic facts, but the work is exclusively on prokaryotic ribosomes. It is important to prevent readers from mistakenly applying knowledge gained here to eukaryotes. The title and abstract should also explicitly state that this is prokaryotic research.

Reply: We agree with the referee that most of the information on the role of m⁶A modification was obtained in eukaryotic systems, whereas the mechanistic work on its effect on decoding was done with *E. coli* translation system. This is justified, because the decoding mechanisms are highly conserved between pro- and eukaryotes, including the structure of the decoding region of the ribosome (Rodnina, 2018, 2023). To avoid the potential confusion, we now indicated which roles of m⁶A are likely eukaryote-specific (e.g., recognition of eIF3, etc, p. 2) and indicate that the mechanistic work was carried out with *E. coli* components (Abstract and p. 2). However, we do think that the reader can apply this knowledge to understand the effects of m⁶A modification on decoding eukaryotes, although the magnitude of the effect may be attenuated by the presence of eukaryotic tRNAs, as indicated in Fig 3 for yeast tRNA^{Lys}.

2. To account for Nature Communication's general audience, the manuscript should be more accessible. The authors should provide more detailed explanations of their approaches, the rationale behind each step, and how to interpret the results, rather than just referring to previous publications.

Reply: Due to space restrictions, we aimed to keep descriptions of the individual parts of our multidisciplinary approaches as concise as possible, although in all cases we provide a short description of the rationale and the method. The rational and experimental approach for ensemble kinetics is explained in detail on pp. 3-4 and illustrated in **Figure 1a**; minor changes introduced on p. 5 are meant to further improve the readability. Also the design of cryo-EM experiments described in great detail (p. 13). In the revised version, we have extended our explanations of the smFRET approach on (p. 8); it is also illustrated in **Figure 4a**.

3. The paper employs a simpler sorting scheme than what was previously published by the Rodnina and Stark labs, which accounted for different ratcheting positions to obtain precise conformational distributions. It is surprising that ratcheting appears to have a minor role here. Moreover, the approach may be biased and not up to the current standards of the ribosome structural field, as some particles result in "low resolution maps." The authors should consider using a more advanced approach.

Reply: We have performed extensive sorting schemes and state-of-the-art data analyses pipelines. For instance, we tested different focused classification regimes and processing pipelines, including masking of the E site and the 30S subunit to determine its rotation state and position. We also performed masked classifications for the EF-Tu region, but we did not detect any EF-Tu signal in the

analyzed samples. In contrast to the studies by the Rodnina/Stark labs, we did not aim at performing time-resolved cryo-EM, but instead analyzed the final reaction product after 60 s of incubation of the ICs with TC prior to vitrification. In agreement with the work on ribosomes from the last two decades, we found a mixture of ribosomes in classical states and in a rotated state with tRNA in hybrid positions (the latter also called ratcheted). As the smFRET data did not show any effect on the transitions from classical to rotated/hybrid states (Table 1), we decided not to direct our analyses on the hybrid/rotated subsets of particles and to focus on the decoding process rather than on the dynamics of pre-translocation complexes. The observed hybrid states produce less well-resolved maps due to the dynamic nature of the A/P tRNA upon ratcheting (P/E A/P to P/E A/P* movement) and the loose coupling between the ribosome rotation and tRNA movement towards the hybrid states (Fischer et al., 2010). If the complementary approaches (smFRET and kinetics) would have indicated that tRNA and ribosome dynamics in the post-decoding state is affected by the m⁶A modification, we would have designed the cryo-EM experiments differently, i.e. in an analogous way to (Carbone et al., 2021; Fischer et al., 2010; Petrychenko et al., 2021). In contrast, our aim for this paper was to obtain high-resolution states that would allow us to visualize and compare the tRNA-mRNA interaction immediately before, during or after decoding. In summary, we still believe that that our strategy was most appropriate for the purpose of the presented work and focused on understanding the role of m⁶A in the decoding process.

We have revised the text to emphasize that map quality is indeed the most important and only appropriate measure for our structural conclusions. We sought to prepare and study the most relevant samples and to obtain maps with high enough resolution to support our scientific claims. Please also see below the detailed answer to the minor issue related to our local resolution maps (raised by the same reviewer).

4. The pi-stacking and conformational change are crucial to the manuscript, but their presentation in the figure is difficult to appreciate. I suggest providing more density, a closer view, and a different viewing angle to better understand and evaluate the claims.

Reply: In response to this comment and to the request related to the same figure by reviewer 2, we have revised **Figure 7**. We now not only show a close-up view for the AAm⁶A IC dataset, but also an additional perspective of the stacking array for both unreacted (IC) structures. The quality of the maps in this region is excellent (2.4-2.7 Å resolution). We now show the density for all stacked bases of the unmodified and modified unreacted ICs in figure 7 – please also see detailed response to reviewer 2. In addition, we have modified and enlarged the display of the decoding regions in Extended data figures 4, 8 and 10.

We would be more than happy to share all our experimental maps and models with the reviewer, in case he/she wants to assess the stacking conformation in greater detail – due to the relatively large size of the map files, we had issues with uploading them in the submission system. The revised figure is shown below.

New Figure 7:

Minor issue:

5. The authors frequently use overall resolutions to argue map quality. However, this can be misleading, as the overall resolution of ribosomes is generally good, particularly for prokaryotic ribosomes. The study discusses local details, so local resolutions should be mentioned in the main text. Extended Data Figure 4 could be improved by enlarging the encircled base densities and reducing the ribosome size to emphasize the relevant features.

Reply: We agree with the reviewer that the overall resolution is not an ideal measure for the map quality in a specific region – especially in maps of a relatively large molecule like the ribosome. We do also believe that providing the local resolution for each mentioned base/region would cause more confusion for the reader. We now mention the local resolution range for the A sites on p. 15 of the revised manuscript. In addition, we have not only enlarged the circles in **Extended Data Figure 4, 8, and 10**, but also show larger portions of the respective A sites. Finally, we also changed the color gradient for the resolution range, as we realized that the color white is a suboptimal choice to label a specific resolution range. We are sorry for the inconvenience and hope that this issue is now satisfactorily addressed.

Please note that the respective order of the Expanded data Figures has been changed to fit the changes in the text.

Again, we would like to highlight that we would be more than happy to share all our experimental maps and models with the reviewer if he/she wants to assess the stacking conformation in greater detail. Due to the relatively large size of the map files, we had issues with uploading them in the submission system.

Reviewer #2 (Remarks to the Author):

This manuscript from Jain S. and colleagues aims at identifying which translocation step is impacted by the presence of a m6A modification. To answer this question the authors used a combination of smFRET, cryoEM and stop-flow rapid kinetics in a fully reconstituted translation system. They identified that m6A has a strong impact on the acceptance rate of the tRNA leading to a quick dissociation of the tRNA.

This is a fascinating use of complementary techniques (smFRET, cryoEM, rapid kinetics) to address this question. This leads to a complete overview of the defects caused by the modification depending on its relative position within the codon.

The experiments have been carefully done, and are very well interpreted, the discussion is very interesting and clearly recapitulate the data obtained during this work.

Reply: We thank the reviewer for positive comments, interesting questions and excellent suggestions to improve the manuscript.

My main concern is about the model they used. Indeed, they used a AAA codon (Lys) and ACC (Thr) to address the question of the role of m6A at the first position of the codon. They found there is a strong difference between both codons (ACC being 8-fold less impacted than AAA by the presence of the modification see fig2B) but they only rely on the literature for the other codons (although their data do not fit with previously published data on the role of m6A) with a A at the second and third position, such as gln (CAG) and pro (CCA).

AAA is a very special codon, as stated by the authors, due to its stacking conformation in absence of tRNA. Although this happens in various organisms like yeast, it is not clear for me if this is true for all codons or not.

Reply: We agree with the reviewer that the quantitative effect of m⁶A depends on the codon, as shown in **Figure 2** and described in the corresponding text. From this point of view, our data are actually consistent with the previous results (Choi et al., 2016; leong et al., 2021), as indicated on p. 6. The differences to earlier reports are in the mechanistic understanding of the effects, which in the previous work was attributed to the impaired association of the TC (leong et al., 2021). This suggestion was based on extrapolations, rather than on direct measurements (as described in the Introduction, p. 2). As we can monitor the association and codon recognition steps directly (**Figure 1** and the respective text), and the rates of these reactions turned out to be independent of the modification, we had to search for other explanations for the reduced GTPase and peptide bond formation rates, which led to experiments in **Figures 1-4**, as stated on p. 8. To address this concern directly, we added a sentence in the Discussion, p. 17.

Concerning AAA being a special codon, the effect of m⁶A seems to be modulated by the context and the position in the codon, rather than by AAA being a special codon (**Figures 1-3**). For example, the effect is somewhat different for Lys2 and Lys4, although this is the same AAA codon (this paper), and

depends on the m⁶A position on the codon (this paper and (Choi et al., 2016)), while the effect of Am⁶AA is similar to that of Cm⁶AG (Choi et al., 2016). This point is discussed on pp. 6 and 17.

Codon stacking of AAA was observed in the *E. coli* (this paper) and yeast (Chandrasekaran et al., 2019; Tesina et al., 2020) systems and was not reported for other codons so far. However, this may be due to the lack of resolution for the A-site mRNA codon in the absence of a bound cognate tRNA, so the question of whether there is stacking of codons other than AAA remains open. This is stated on p. 17.

This leads me to question the generalizability of these data. This is not really discussed in the manuscript, and I feel this could be a real improvement to perform the kinetics analysis with such codons. This is not a negligible question, because m⁶A is mainly found in coding sequences, so if m⁶A has a so strong effect on translation, it is very surprising that this does not lead to more translational defects, especially in cancer cells where m⁶A homeostasis is strongly modified.

Reply: The kinetic analysis presented in **Figures 1-3** and smFRET in **Figure 4** require fluorescence-labeled tRNAs that have to be well characterized to show that their activities are not changed by fluorescence labeling. The tRNAs used in this study, tRNA^{Lys} and tRNA^{Phe} can be labeled and their activity was extensively tested and validated by previous work (Adio et al., 2015; Gromadski and Rodnina, 2004; Pape et al., 1999; Pape et al., 1998). Unfortunately, due to the lack of fluorescent derivatives of tRNAs reading other codons (Glu, Gln, Pro, and Thr), it is not possible to perform the same sets of experiments that we did with rapid kinetics and smFRET approaches for tRNA^{Lys}. For these other tRNAs the readout is indirect, i.e., limited to GTPase/peptide bond formation and ribosome rotation, as was already carried out in (Choi et al., 2016) and (leong et al., 2021), and which does not allow to measure the discard rates of tRNAs. In addition, the choice of the AAA codon allowed us to systematically study the effect of the modification in all three codon positions (p. 3), which would be not possible with any other codon.

Concerning further translational defects by m⁶A, we have studied the effect of the modification on tRNA-mRNA translocation (2-fold effect; Extended Data Fig. 1b). We have shown that post-decoding complexes do not differ either in structural characteristics (as shown in the cryo-EM section) or rates of interconversion between classical and hybrid states (shown in the smFRET data). Recent study reported that m⁶A decreases the stop codon reading efficiency by RF2, which would affect translational termination (leong et al., 2021). Thus, the effects of the m⁶A modification on the translation of coding sequences are quite complete.

We agree with the referee that understanding the cellular and organismic effect of m⁶A, in particular in cancer where m⁶A homeostasis is modified, is an outstanding question. However, answering this question would require a totally different set of experiments, such as ribosome profiling and proteome/nascentome analysis by mass spectrometry to look for potential ribosome pausing, dysregulation of protein folding, and the role of quality control systems (chaperones, protein degradation) on protein homeostasis. These experiments are entirely beyond the scope of the present paper, which focuses on the molecular mechanism of the effect.

I have few more comments for the authors:

- It would help the reader to better understand the experimental setup if the mRNA sequence used was indicated in the various figures. This is especially true for the context effect, where two lysine codons are positioned at position 2 and 4 of the mRNA. Do the differences observed between both codons due to the nucleotide located immediately downstream? This position effect is very curious, and the authors do not comment on that.

Reply: We added the mRNA sequence to **Fig 2a and b**.

The mRNA context of Lys2 and Lys4 is different, but also the timing of decoding changes due to translation of codons 2 and 3 prior to reading the Lys4 codon. We note that the results of the experiment are interpreted solely to indicate that the m⁶A effect can be measured at different codon positions in the mRNA. The exact quantification of L2-L4 effect (and thus any conclusions with respect to potential effects of nucleotides upstream and downstream of the codon) is difficult due to the translation delay prior to Lys4 incorporation. The 5' nucleotide of the AAA and ACC codons is the same.

- In the first experiments it is not clear if the lysine tRNA used is fully modified or not. As this is a fully reconstituted translation system, it will be useful to clearly indicate modification status of the tRNA. Do the authors verify that after the isolation and the labelling tRNAs were still modified?

Reply: The *E. coli* Lys-tRNA^{Lys} and yeast Phe-tRNA^{Phe} was prepared from the respective cells according to previously published protocols (Milon et al., 2007; Ranjan and Rodnina, 2017). Lys-tRNA^{Lys} that lacks modifications is inactive in decoding, whereas yeast tRNA^{Phe} is active also without the modifications. Notably, we did not use tRNA transcripts. Purification and labeling of the tRNA has no effect on its modification status (Pape et al., 1998; Ranjan and Rodnina, 2017).

- Figure 7 is not completely clear. I guess we are looking ribosomes with an empty A site as suggested by the panel A? So why the panel D shows a post-decoding structure with a A-site tRNA?

Reply: We have rearranged **Figure 7** in the revised manuscript to clarify this point. We initially aimed to provide a direct structural comparison between the A site of the "unreacted IC" (**Figure 7c**) and the P/P A/A state of the complex in the absence of m⁶A modification (old **Figure 7d** and also described in **Figure 6a**). Considering the comment by the reviewer, we now realized that the old figure panel 7d is indeed misplaced and partially redundant with the old panel 7e. As we still want to highlight the quality of the map for the monitoring bases (see minor issue by reviewer 1), we moved a modified version of **Figure 7d** to the **Extended Data Figure 12**.

In addition, we would like to emphasize that in all structures the AAA codon was in a π -stacking conformation. To illustrate this, we now incorporated a new figure panel in **Figure 7** that shows the identical close-up view (as in **Figure 7c**) for the AAm⁶A from the IC dataset. In addition, we added a top-view to both panels, showing the stacking of the monitoring and codon bases (see response to issue #4 of reviewer 1). **Figure 7d** remains identical, still showing the intended comparison between IC and the P/P A/A structure and the specific movement of the monitoring bases.

We updated the respective figure legends and the references to the figures in the main text.

Reviewer #3 (Remarks to the Author):

This manuscript from a leading laboratory in studies of protein synthesis mechanisms contains several interesting results well worthy of publication, combining ensemble, single molecule, and cryoEM studies in attempting to get a complete picture of the effects of m⁶A mRNA modification on translation. However, it is poorly presented and should be rejected in its current form.

Reply: We thank the reviewer for the comments on the manuscript.

MAJOR CRITICISMS

1. There is a consistent lack of rigorous, quantitative reasoning, which detracts from the significance of the results presented. In principle, the ensemble and single molecule results should reinforce one another, both qualitatively and quantitatively, but such direct comparisons are lacking. As one example, there is close to 100% of product formed for the AAA and m⁶AAA ensemble experiments (Fig. 2), but only more partial formations in the single molecule experiments (Fig. 4b) and this apparent discrepancy is not explained, nor are the extremely low fractions of product formation found in the cryo-EM experiments (Fig. 6a).

Reply: We regret that the reviewer did not appreciate our efforts to combine three different approaches in a coherent picture, while at the same time being careful not to overinterpret data that are by necessity obtained in different experimental setups. Specifically, the apparent discrepancy in the end level of the ensemble and smFRET/cryo-EM data results from the simple fact that in **Figures 2A** and **Extended Data Figure 1**, the end levels are normalized to allow for the direct comparison of rates, as is the standard practice in the field (Choi et al., 2016; Jeong et al., 2021). In contrast, smFRET and cryo-EM data provide an inventory of the complexes, however at their respective concentrations, which are mandated by the technical specifics of the respective approach. Thus, making direct quantitative comparisons between the methods would not be justified. Instead, such comparisons must be done in a qualitative way considering the specific experimental design of each approach. Thus, we strongly disagree with the referee on this point.

Other examples of a lack of rigorous, quantitative reasoning:

p.6 – para 1 - penultimate sentence. What is the actual estimated delay for Lys 4 , and how was it quantified?

Reply: The estimated delay in decoding due to m⁶A modification was calculated in the following way. The reaction time τ calculated from exponential fitting is 0.08 s for AAA vs 3.3 s for m⁶AA at Lys2 and 2 s for AAA vs 18 s for m⁶AAA at Lys4. Thus, the delay in decoding due to the modification is ~ 3 s (3.3-0.8) at Lys2 and ~ 16 s (18-2) for Lys4. This information is now added to **Figure 2**, legend.

p. 7 – lines 5 – 8. The biphasic pathway. How was it shown that Lys-tRNA was fully modified? What other explanations of heterogeneity are possible? Why are m⁶AAA and AAm⁶A apparently monophasic.

Reply: The origin of the biphasic pathway has been extensively studied in (Ranjan and Rodnina, 2017), including a full kinetic analysis of the reaction with alternative pathways and global fitting over eight different observables (Ranjan and Rodnina, 2017). The biphasic reaction is due to alternative pathways of tRNA accommodation, which depend on the timing of tRNA release from EF-Tu (Ranjan and Rodnina, 2017). As shown in that paper, the biphasic pathway is not due to heterogeneity of the tRNA, because other reactions, e.g. GTP hydrolysis, are clearly monophasic, underscoring the homogeneity of the TC. The tRNA used in that and the present paper is prepared from yeast strains with or without the specific s² modification enzyme as detailed on p. 7. The extent of s² modification was verified by ((N-acryloylamino)phenyl)mercuric chloride (APM) gel retardation of tRNA and was 70-90% for native tRNA^{Lys} and 2% for *urm1Δ* tRNA^{Lys}; this information is now added in Methods, p. 20. m⁶A modification

reduces the rate of peptide bond formation at a step preceding the tRNA accommodation (this paper). Thus, the rate-limiting step is shifted from the accommodation to an earlier step, which masks any potential biphasic behavior at the subsequent steps.

p. 16 – lines 8, 9 from bottom – Is the energy difference observed in the NMR study sufficient to account for the differences observed in this manuscript?

Reply: Previous study shows that unpaired m⁶A nucleobase prefers the *syn* conformation. However, for m⁶A-U base pairing, N⁶ requires flipping and trapping the methylamino group in energetically unfavorable *anti*- conformation. This is partially compensated for by hydrogen bonding and Watson-Crick pairing with U resulting in the net destabilization by 0.5–1.7 kcal/mol for a single methyl substitution. Studies of 6-methyldeoxyadenosine in duplex DNA suggest that the methyl group slows hybridization, consistent with a requirement for rotation of the methylamino group into a high-energy conformation before base pairing can occur (Roost et al., 2015). The authors of that paper conclude that, when paired in RNA with stable base pairing surrounding it, m⁶A acts as a compressed spring that is locked into place by its paired context (Roost et al., 2015). This observation matches our results where we show that the methyl group is highly flexible and cannot be captured despite the high resolution of the nucleotide base they are attached to.

2. Differences are cited with the work of Puglisi et al., but no attempt is made to explain possible reasons for these differences.

Reply: Our results are very similar to the data of Puglisi et al (Choi et al., 2016) in the magnitude of the m⁶A effect on GTP hydrolysis and peptide bond formation and provide further information on the codon position and context effects that were not directly tested by Choi et al. This is now stated more clearly on p. 17. The difference to the previous reports is in the mechanism by which m⁶A effect is exerted. Specifically, as stated in (Jeong et al., 2021), their model suggests that the association of TC with the ribosome is affected (see Introduction, p. 2). However, that model was based on the extrapolations of the k_{cat}/K_M values, which provide a lower limit to the association rate constant, but do not measure the k_a directly. In contrast, we measure the association and codon recognition rates directly (**Figure 1** and the respective text) and show that there are no differences with or without the modification. Our smFRET and cryo-EM analysis provides an alternative mechanism for the m⁶A effect.

3. UNJUSTIFIED CONCLUSIONS.

p. 18 – lines 1-5. The conclusion that the difference in mcm5s2U34 vs. mnm5s2U34 in yeast vs. E. coli tRNAs accounts for the effects on peptide bond formation is totally unjustified, since other differences in sequence and modification of the two tRNAs (lines 1 and 2) could also be important. Indeed, the comments about biphasic kinetics that follow, lines 3-5, invoke such differences.

Reply: We agree with the referee and removed the respective sentences on p. 19. Our intention was really to compare tRNA^{Lys} with the fully modified mcm⁵s²U₃₄ to hypomodified mcm⁵U₃₄ tRNA^{Lys} (Fig 3 and the respective text on pp. 7 and 19).

p. 17 – lines 3 – 5 from the bottom. Ribosome rectification of codon-anticodon dynamics. This conclusion is more strongly worded than is justified by the results. What other possibilities were considered?

Reply: Our suggestion is based on the results of cryo-EM as argued on pp. 17-18. Despite the high local resolution of the structures, the methyl group remains dynamic prior to tRNA binding, most likely owing to its high flexibility. Upon A-site tRNA binding, the N⁶ group is stabilized and locked in

energetically unfavorable *anti*- conformation. In this conformation, the Watson-Crick geometry of the codon-anticodon pair is similar for unmodified and m⁶A modified AAA codon. The methyl group in these complexes can also be mapped in *syn*- confirmation, however this increases the distance of the N⁶ atom to the O4 atom from the cognate U in the codon by approximately 0.5 Å. Such a base pair would not be energetically favorable. We thus conclude that the ribosome selects for the correct conformation of mRNA codon or it rectifies the codon conformation upon allowing tRNA binding. Previous work on ribosomes also argues in favor of rectification of codon-anticodon dynamics, e.g., selection by the ribosome of a rare tautomeric form of the G-U base pair results in misreading of codons by near-cognate tRNAs (Ogle and Ramakrishnan, 2005; Rozov et al., 2018). A sentence in this direction has been added on p. 19.

OTHER CONCERNS

1. p. 4, Fig 1d – There is an apparent error in the results presented for AAA, since tau GTPase must be > tau peptide.

Reply: As GTP hydrolysis precedes peptide bond formation, the reaction time for GTP hydrolysis (τ_{GTP} , s) must be smaller than that for peptide bond formation (τ_{pep} , s). For unmodified AAA codon, the two values are similar within the standard deviation, as peptide bond formation occurs rapidly after GTP hydrolysis. In contrast, in the presence of m⁶A, both GTP hydrolysis and subsequent peptide bond formation steps are slower, delaying the time taken by the ribosomes to complete individual kinetic steps.

2. p. 14 –The last sentence appears to be in direct contradiction of the major thesis of the manuscript.

Reply: We do not see a contradiction, but altered the sentence to avoid potential misunderstanding.

3. The ensemble rate data, Figs. 1-3, should be summarized in a Table, for easy reference by the reader.

Reply: All rates are provided in the respective Fig legends for easy reference by the reader. Providing an additional Table would be duplicating this information, which we would prefer to avoid.

4. Rate units should be used consistently for both ensemble and sm data. The switch within the MS between k (s⁻¹) and tau(s), as in Fig. 1, is needlessly confusing.

Reply: We specifically explain on p. 5 that we switch from rates to reaction rates τ , because it is intuitively easier to understand the differences (the rates of the preceding reactions are very similar, so this does not apply to those). Using τ has advantages for calculating delay times, such as in Fig 2a. In all other cases we use rates (see Table 1).

MINOR POINT

p.19 – line 3. The tRNAs should be identified as coming from E. coli.

Reply: tRNA sources are updated accordingly in the Material and Method section of the manuscript (p. 20).

References

- Adio, S., Senyushkina, T., Peske, F., Fischer, N., Wintermeyer, W., and Rodnina, M.V. (2015). Fluctuations between multiple EF-G-induced chimeric tRNA states during translocation on the ribosome. *Nat Commun* 6, 7442.
- Carbone, C.E., Loveland, A.B., Gamper, H.B., Jr., Hou, Y.M., Demo, G., and Korostelev, A.A. (2021). Time-resolved cryo-EM visualizes ribosomal translocation with EF-G and GTP. *Nat Commun* 12, 7236.
- Chandrasekaran, V., Juskiewicz, S., Choi, J., Puglisi, J.D., Brown, A., Shao, S., Ramakrishnan, V., and Hegde, R.S. (2019). Mechanism of ribosome stalling during translation of a poly(A) tail. *Nat Struct Mol Biol* 26, 1132-1140.
- Choi, J., leong, K.W., Demirci, H., Chen, J., Petrov, A., Prabhakar, A., O'Leary, S.E., Dominissini, D., Rechavi, G., Soltis, S.M., et al. (2016). N(6)-methyladenosine in mRNA disrupts tRNA selection and translation-elongation dynamics. *Nat Struct Mol Biol* 23, 110-115.
- Fischer, N., Konevega, A.L., Wintermeyer, W., Rodnina, M.V., and Stark, H. (2010). Ribosome dynamics and tRNA movement by time-resolved electron cryomicroscopy. *Nature* 466, 329-333.
- Gromadski, K.B., and Rodnina, M.V. (2004). Kinetic determinants of high-fidelity tRNA discrimination on the ribosome. *Mol Cell* 13, 191-200.
- leong, K.W., Indrisiunaite, G., Prabhakar, A., Puglisi, J.D., and Ehrenberg, M. (2021). N 6-Methyladenosines in mRNAs reduce the accuracy of codon reading by transfer RNAs and peptide release factors. *Nucleic Acids Res.*
- Milon, P., Konevega, A.L., Peske, F., Fabbretti, A., Gualerzi, C.O., and Rodnina, M.V. (2007). Transient kinetics, fluorescence, and FRET in studies of initiation of translation in bacteria. *Methods Enzymol* 430, 1-30.
- Ogle, J.M., and Ramakrishnan, V. (2005). Structural insights into translational fidelity. *Annu Rev Biochem* 74, 129-177.
- Pape, T., Wintermeyer, W., and Rodnina, M. (1999). Induced fit in initial selection and proofreading of aminoacyl-tRNA on the ribosome. *EMBO J* 18, 3800-3807.
- Pape, T., Wintermeyer, W., and Rodnina, M.V. (1998). Complete kinetic mechanism of elongation factor Tu-dependent binding of aminoacyl-tRNA to the A site of the E. coli ribosome. *EMBO J* 17, 7490-7497.
- Petrychenko, V., Peng, B.Z., de, A.P.S.A.C., Peske, F., Rodnina, M.V., and Fischer, N. (2021). Structural mechanism of GTPase-powered ribosome-tRNA movement. *Nat Commun* 12, 5933.
- Ranjan, N., and Rodnina, M.V. (2017). Thio-Modification of tRNA at the Wobble Position as Regulator of the Kinetics of Decoding and Translocation on the Ribosome. *J Am Chem Soc* 139, 5857-5864.
- Rodnina, M.V. (2018). Translation in Prokaryotes. *Cold Spring Harb Perspect Biol* 10.
- Rodnina, M.V. (2023). Decoding and Recoding of mRNA Sequences by the Ribosome. *Annu Rev Biophys* 52, 161-182.
- Roost, C., Lynch, S.R., Batista, P.J., Qu, K., Chang, H.Y., and Kool, E.T. (2015). Structure and thermodynamics of N6-methyladenosine in RNA: a spring-loaded base modification. *J Am Chem Soc* 137, 2107-2115.
- Rozov, A., Wolff, P., Grosjean, H., Yusupov, M., Yusupova, G., and Westhof, E. (2018). Tautomeric G*U pairs within the molecular ribosomal grip and fidelity of decoding in bacteria. *Nucleic Acids Res* 46, 7425-7435.
- Tesina, P., Lessen, L.N., Buschauer, R., Cheng, J., Wu, C.C., Berninghausen, O., Buskirk, A.R., Becker, T., Beckmann, R., and Green, R. (2020). Molecular mechanism of translational stalling by inhibitory codon combinations and poly(A) tracts. *EMBO J* 39, e103365.

REVIEWERS' COMMENTS

Reviewer #1 (Remarks to the Author):

I thank the authors for providing the maps and they look indeed grate. With this the authors have addressed my concerns sufficiently and I recommend publication.

Reviewer #2 (Remarks to the Author):

The authors have responded convincingly to all my comments. In particular, I find the new Figure 7 much clearer.

This is a truly original and interesting work that uses cutting-edge approaches to answer an important question about the role of m6A modifications present in coding sequences on the ribosome.

I would like to congratulate the authors on the quality of their experiments and their careful approach to analyzing the results.

Reviewer #3 (Remarks to the Author):

The authors have responded positively to some of my earlier criticisms, but I still would not recommend acceptance of the manuscript in its present form. My major objection is that the authors have not convincingly demonstrated that the three sets of experiments presented together yield a consistent picture of the effects on translation of m6A modification.

Two points require further revision.

1. Given the extensive concentration dependent results presented in the ensemble experiments, it should be possible to directly compare single molecule and ensemble rate results, as other investigators have done. With respect to stoichiometries, while it is OK to present normalized results, the relevant Figure legend should make that point explicitly and the normalization factor should be provided (neither is done in the revised manuscript), so that stoichiometries are clear to the reader and can be compared between the ensemble and single molecule results.

2. I agree that the cryoEM results cannot be quantitatively compared with the single molecule and ensemble results. However, even on a qualitative basis I found it somewhat surprising that, lines 411-4 “the positions of these rRNA and mRNA nucleotides are identical in the m6A-modified and the AAA datasets (Fig. 7c), showing that m6A does not affect monitoring of the codon-anticodon complex by the ribosome.” Why then is there a 40-fold difference in the rate of peptide formation? This question needs to be addressed by the authors.

In addition, convincing rebuttals were made to questions raised in my earlier review which should be included in a final text so that they will be available to the readers, but are not present in the current revision.

1. Differences with Choi et al., 2016. The 40-fold effect on peptide bond formation of m6A modification (p. 6) is much larger than that found in Choi et al 2016. The explanation for this difference contained in the rebuttal remarks to me and to reviewer 2 should be included.
2. The biphasic pathway. original p. 7 – lines 5 – 8. The quantitative values of modification should be included, in addition to a brief discussion of the evidence for a biphasic the biphasic pathway.
3. The consistency with a prior NMR study. original p. 16 – lines 8, 9 from bottom – the cited destabilization energy and the Roost et al. conclusion are pertinent and should be included.

Reply to Reviewer #3

Two points require further revision.

1. Given the extensive concentration dependent results presented in the ensemble experiments, it should be possible to directly compare single molecule and ensemble rate results, as other investigators have done. With respect to stoichiometries, while it is OK to present normalized results, the relevant Figure legend should make that point explicitly and the normalization factor should be provided (neither is done in the revised manuscript), so that stoichiometries are clear to the reader and can be compared between the ensemble and single molecule results.

Reply: We politely disagree with the reviewer concerning the direct quantitative comparison of single molecule and biochemical data. The quantitative comparisons suggested by the referee may be possible in some specific cases where the experiments can be done at exactly the same conditions. However, in the present case, ensemble kinetics and smFRET experiments have to be carried out at grossly different concentrations of ribosomes and ternary complexes and at somewhat different buffer conditions, which presents a fundamental obstacle for such direct comparisons.

We have added the statement that the time courses for GTPase and dipeptide were normalized from 0 to 1, and used this occasion to explain how we calculate relative fluorescence in Figs 1b,c (Methods, p. 15). We refrained from giving normalization factors (the normalization is done automatically by GraphPrism), because this value is potentially misleading and it is difficult to see what could be gained by quantifying this factor. One could think that the reaction end-levels could be used to determine the drop-off rate of the ternary complex/tRNA. However, in this particular case, ternary complexes rejected in the first round can re-bind and make multiple attempts to accommodate, meaning that the final end-level of ensemble measurements is a mixture of the 1st and multiple reaction rounds. Extracting the k_{off} from these experiments is essentially not possible due to the multistep binding mechanism. This is not the case in the smFRET experiment, where every reaction is distinguishable and the k_{off} can be reliably calculated.

We would like to emphasize that the way we carry out the experiments and evaluate the data follow established standards in the field. For example, (Choi et al., 2016; Jeong et al., 2021) also do not report exact stoichiometries and present normalized ensemble kinetics data. In our paper, the information on the drop-off is readily available from smFRET and cryo-EM inventories. In summary, for the present experimental goals it is not feasible, but also not necessary, to have quantitative direct comparison of ensemble and smFRET data and it would not be justified (and even misleading) to report the values suggested by the referee.

2. I agree that the cryoEM results cannot be quantitatively compared with the single molecule and ensemble results. However, even on a qualitative basis I found it somewhat surprising that, lines 411-4 “the positions of these rRNA and mRNA nucleotides are identical in the m6A-modified and the AAA datasets (Fig. 7c), showing that m6A does not affect monitoring of the codon-anticodon complex by the ribosome.” Why then is there a 40-fold difference in the rate of peptide formation? This question needs to be addressed by the authors.

Reply: The sentence addressed by the referee describes the conformation of mRNA in the initiation complex (IC) prior to ternary complex binding (last sentence of the Results). In the Discussion, which directly follows, we explain why the rates of decoding reactions are changed and in particular emphasize that the modification affects the dynamics of the codon-anticodon complex, including stabilization of poly(A) array stacking (p. 11) and preference for syn vs. anti conformation (p. 12). The ribosome selection of the codon-anticodon geometry is described on pp. 18-19. The question raised

by the referee is addressed already in the Abstract: “m⁶A does not exclude canonical codon-anticodon geometry, but favors alternative more dynamic conformations that are rejected by the ribosome” and then on pp. 11-13 of the Discussion.

In addition, convincing rebuttals were made to questions raised in my earlier review which should be included in a final text so that they will be available to the readers, but are not present in the current revision.

1. Differences with Choi et al., 2016. The 40-fold effect on peptide bond formation of m⁶A modification (p. 6) is much larger than that found in Choi et al 2016. The explanation for this difference contained in the rebuttal remarks to me and to reviewer 2 should be included.

Reply: Choi et al presents two types of experiments that we can use for comparison. One is based on the subunit rotation smFRET experiment (Fig. 2 in Choi et al), which is a proxy for peptide bond formation, because subunits rotation requires and rapidly follows peptide bond formation. Choi et al study AAA codon at position 4 (K4 in their nomenclature), which corresponds to our Lys4 in Fig. 2 (note that Lys1, Lys2, Lys3 in Choi denote the position of modification in the codon). Fig 2c in Choi et al shows the 15-fold effect of m⁶AAA, which is consistent with our data (10-fold for the same position); this statement is now added on p. 5. They also present GTPase and dipeptide experiment for Lys codon at position 2 using ensemble kinetics and report a 10-fold effect by m⁶AAA, which is also very similar to our effect of 8-fold (p. 4). In contrast, dipeptide formation in Choi et al (Fig 3f) appears to be affected only about 2-fold. However, these values represent a calculated transit time between the successful GTPase and dipeptide formation, which is not the same as the measured dipeptide rates presented in our Fig 1. Calculating the transit times reduces the potential difference, because these data are carried out on normalized data (Fig. 3e,f in Choi et al.) and disregard potential differences in the end-levels due to drop-off. This is clearly not justified, e.g. comparison of the dipeptide rates and ribosome rotation rates in Choi immediately shows that the values are not comparable, as there is a 10-fold difference between them, notably in the same paper). This is why we do not make such calculations and instead present the measured data in a model-independent way.

In summary, ribosome rotation and GTPase experiments are not that different in our and Choi experiments, which is stated in several places of the manuscript. The major difference in interpretation is described at the beginning of the Discussion (p. 17; added upon previous revision round).

2. The biphasic pathway. original p. 7 – lines 5 – 8. The quantitative values of modification should be included, in addition to a brief discussion of the evidence for a biphasic the biphasic pathway.

Reply: we added the quantification as requested in Materials (p. 20) and a reference to Materials in the text (p. 6). The discussion of the biphasic pathway for the hypomodified mcm⁵s² Lys-tRNA^{Lys} is beyond the scope of this paper, because it would disrupt the text flow and because all these arguments and very detailed quantification of the pathway are presented in ref. 22, which deals exactly with the kinetic mechanism of fully modified and hypomodified tRNA^{Lys}. This earlier careful work required considering different pathways and is based on extensive kinetic analysis; it is simply not possible to repeat all these arguments here.

3. The consistency with a prior NMR study. original p. 16 – lines 8, 9 from bottom – the cited destabilization energy and the Roost et al. conclusion are pertinent and should be included.

Reply: we have added the relevant sentence on p. 12.